# Score-based generative models break the curse of dimensionality in learning a family of sub-Gaussian probability distributions

**Frank Cole**
Department of Mathematics
University of Minnesota
Minneapolis, MN, 55414, USA
`{cole0932}@umn.edu`

**Yulong Lu**
Department of Mathematics
University of Minnesota
Minneapolis, MN, 55414, USA
`{yulonglu}@umn.edu`

## ABSTRACT

While score-based generative models (SGMs) have achieved remarkable successes in enormous image generation tasks, their mathematical foundations are still limited. In this paper, we analyze the approximation and generalization of SGMs in learning a family of sub-Gaussian probability distributions. We introduce a notion of complexity for probability distributions in terms of their relative density with respect to the standard Gaussian measure. We prove that if the log-relative density can be locally approximated by a neural network whose parameters can be suitably bounded, then the distribution generated by empirical score matching approximates the target distribution in total variation with a dimension-independent rate. We illustrate our theory through examples, which include certain mixtures of Gaussians. An essential ingredient of our proof is to derive a dimension-free deep neural network approximation rate for the true score function associated to the forward process, which is interesting in its own right.

## 1 INTRODUCTION

Generative modeling is a central task in modern machine learning, where the goal is to learn a high dimensional probability distribution given a finite number of samples. Score-based generative models (SGMs) Sohl-Dickstein et al. (2015); Song et al. (2021)) recently arise as a novel family of generative models achieving remarkable empirical success in the generation of audio and images Yang et al. (2022); Croitoru et al. (2023), even outperforming state-of-the-art generative models such as generative adversarial networks Dhariwal and Nichol (2021). More recently, SGMs have proven effective in a variety of applications such as natural language processing Austin et al. (2021); Savinov et al. (2021), computational physics Lee et al. (2023a); Jing et al. (2022), computer vision Amit et al. (2021); Baranchuk et al. (2021); Brempong et al. (2022), and medical imaging Chung and Ye (2022). In addition to their own expressive power, SGMs can also help to understand and improve other existing generative models, such as variational autoencoders Huang et al. (2021); Luo (2022) and normalizing flows Gong and Li (2021)).

SGMs are often implemented by a pair of diffusion processes, known as forward and backward processes. The forward process transforms given data into pure Gaussian noise, while the backward process turns the noises into approximate samples from the target distribution, thereby accomplishing generative modeling. The analytical form of the reverse process is unknown, since its parameters depend on the target distribution, which is only accessible through data; hence, the reverse process must be *learned*. This is made possible by the remarkable fact that the time reversal of an diffusion process is again a diffusion process whose coefficients depend only on the target distribution via the *score function*, a time-dependent vector field given by the gradient of the log-density of the forward

process. There exist well-studied techniques to cast the estimation of the score function from data as a supervised learning problem Hyvärinen and Dayan (2005); Vincent (2011), which is crucial to the practical implementation of SGMs.

While SGMs have received significant attention from theoretical viewpoints, there are still several barriers to a complete theoretical understanding. Recent results Chen et al. (2023a;b); Benton et al. (2023) have shown that the distribution recovery error of SGMs is essentially controlled by the estimation error of the score function, which is typically parameterized by a neural network. While neural networks are known to be universal approximators for many classes of functions Cybenko (1989); Yarotsky (2017), the number of parameters of the neural network needed to approximate a function to error $\epsilon$ often scales like $\epsilon^{-d}$, where $d$ is the dimension of the data. Such rates are of little practical significance for high dimensional problems, and thus the ability of neural networks to express the score function of a general probability distribution remains a mystery.

Nonetheless, SGMs have still exhibited great success in generating high-quality samples from complex, high-dimensional data distributions. One salient reason for this is that, while the data itself may be very high-dimensional, the score function of the noising process often possesses some intrinsic structure that can be exploited by neural networks. The purpose of this article is to justify this intuition rigorously for a broad class of probability distributions. Specifically, we study the generative power of SGMs for probability distributions which are absolutely continuous with respect to the standard Gaussian distribution. Such distributions admit a probability density function of the form

$$p(x) = \frac{1}{Z} \exp\left(-\frac{\|x\|^2}{2} + f(x)\right), \tag{1}$$

where $\exp(f) : \mathbb{R}^d \to \mathbb{R}^+$ is the Radon-Nikodym derivative of $p$ with respect to the Gaussian distribution and $Z$ is the normalization constant. This representation of the density has proven particularly elucidating in the context of statistical learning and Bayesian inference, where the Gaussian component can model our subjective beliefs on the data. In this paper, we show that the expression for the density in Equation 1 is also relevant to SGMs, because the score function is related to the function $f$ by a tractable composition of functions. A central theme of this work is that if $f$ belongs to a low-complexity function class, then the score function inherits a similar low-complexity structure. This enables deep neural networks to learn the score function of diffusion processes without the curse of dimensionality in some concrete cases.

## 1.1 OUR CONTRIBUTIONS

We summarize our contributions as follows.

1. We prove that if the log-relative density of the data distribution with respect to the standard Gaussian can be locally approximated without the curse of dimensionality, then the score function at any fixed time $t$ can be approximated in the $L^2(p_t)$ norm, where $p_t$ denotes the marginal density of the forward process at time $t$, without the curse of dimensionality.

2. We show that the empirical score matching estimator within a prescribed class neural networks can estimate the score at any fixed time without the curse of dimensionality. The error is decomposed into the approximation error of the score and the Rademacher complexity of the neural network class.

3. We combine our results with existing discretization error bounds (e.g., in Chen et al. (2023b)) to obtain explicit error estimates for SGMs in terms of the number of training samples. As an application, we prove that SGMs can sample from certain Gaussian mixture distributions with dimension-independent sample complexity.

## 1.2 RELATED WORK

A majority of the recent theoretical analysis of SGMs De Bortoli et al. (2021); Lee et al. (2022; 2023b); Chen et al. (2023a;c); Benton et al. (2023) focuses on obtaining convergence guarantees for SGMs under minimal assumptions on the target distribution but, crucially, under the assumption that the score estimator is accurate in the sense of $L^2$ or $L^\infty$. The common message shared among these works is that learning the distribution is as easy (or hard) as learning the score function. More precisely, the estimation error of the target density is mainly controlled by the estimation error of

the score function and the discretization error of the diffusion processes (as another error source) scales at most polynomially in the data dimension. However, there has been relatively little work to address the problem of score estimation error.

More recently, it was proven in Oko et al. (2023) that SGMs can estimate Besov densities with a minimax optimal rate under the total variation distance. However, the obtained sample complexity of density estimation over a Besov space suffers from the curse of dimensionality. The paper Oko et al. (2023) further proved that the estimation rate can be substantially improved under the additional assumption that the data distribution is supported on a low-dimensional linear subspace, in which case the resulting rate only depends on the intrinsic dimension. Distributions with the same low-dimensional structure was also studied by Chen et al. (2023d) in the Lipschitz continuous setting. The paper De Bortoli (2022) obtained convergence bounds in the Wasserstein distance for SGMs under a more general manifold hypothesis on the target distributions (including empirical measures).

Our work differs from the work above in that we do not make low-dimensional assumption on the data distribution. Instead, we assume that the target is absolutely continuous with respect to a Gaussian and that the log-relative-density belongs to the Barron space Barron (1993). Barron functions have recently received much attention due to the fact that shallow networks can approximation them without curse of dimensionality; see, e.g., Klusowski and Barron (2018); Siegel and Xu (2022); Ma et al. (2022). In the context of generative modeling, the recent work Domingo-Enrich et al. (2021a;b) investigated the statistical guarantees of energy-based models under the assumption that the underlying energy function lie in Barron space (or the $\mathcal{F}_1$ space therein). The work Lee et al. (2017) obtained expressive bounds for normalizing flows in representing distributions that are push-forwards of a base distribution through compositions of Barron functions. This work shares the same spirit as Domingo-Enrich et al. (2021a;b); Lee et al. (2017) and demonstrates the statistical benefits of SGMs when target distribution exhibits low-complexity structures.

We note that in an earlier version of this work, the log-relative density $f$ was assumed to be bounded. After discussions with an anonymous reviewer, we were inspired to strengthen the results to allow $f$ to grow at infinity. In more detail, the reviewer pointed out that when $f$ is bounded, the data distribution satisfies a log-Sobolev inequality (LSI) with constant $e^{\|f\|_\infty}$, which implies that the distribution can be sampled via Langevin dynamics with an estimator for the vanilla score. Our current results extend beyond the LSI case.

## 1.3 Notation

Throughout this article, we study functions and probability distributions on a Euclidean space $\mathbb{R}^d$ of a fixed dimension $d$. We let $\|\cdot\|$ denote the Euclidean norm on $\mathbb{R}^d$. For a vector or function, $\|\cdot\|_\infty$ denotes the supremum norm, and $\|\cdot\|_{Lip}$ denotes the Lipschitz seminorm of a metric space-valued function. We let $\gamma_d(dx)$ denote the standard Gaussian measure on $\mathbb{R}^d$, i.e., $\int f(x)\gamma_d(dx) = (2\pi)^{-d/2} \int_{\mathbb{R}^d} f(x)e^{-\|x\|^2/2}dx$. The indicator of a set $S$ is denoted $\mathbb{I}_S$. We denote by $(\cdot)^+$ the ReLU activation function, defined by $(c)^+ = \max(0, c)$ for $c \in \mathbb{R}$. For a vector $x$, $(x)^+$ is interpreted componentwise. For $X_0 \in \mathbb{R}^d$ and $t > 0$, we define $\Psi_t(\cdot|X_0)$ as the Gaussian density function with mean $e^{-t}X_0$ and variance $1 - e^{-2t}$. For non-negative functions $g(x), h(x)$ defined on $\mathbb{R}^d$, we write $g(x) \lesssim h(x)$ (resp. $\gtrsim$) or $g(x) = O(h(x))$ (resp. $\Omega(h(x))$) if there exists a constant $C_d > 0$ which depends at most polynomially on the dimension $d$ such that $g(x) \leq C_d h(x)$ (resp. $\geq$). We write $g(x) = \Theta(h(x))$ if $g(x) \lesssim h(x) \lesssim g(x)$. For $\beta \in (0, 1/2)$ and $g \in L^2(\gamma_d)$ we define $M_\beta(g) := \int_{\mathbb{R}^d} \left| g\left(\frac{x}{1-2\beta}\right)\right|^2 \gamma_d(du)$.

## 2 Background

In this section, we give a brief overview of the mathematical preliminaries required to understand our main result.

## 2.1 A primer on SGMs

In a score-based generative model (SGM), data is first transformed to noise via a forward process; we work with an Ornstein-Uhlenbeck process $(X_t)_{0 \leq t \leq T}$, which solves the stochastic differential

equation

$$dX_t = -X_t dt + \sqrt{2}dW_t, \ X_0 \sim p_0. \tag{2}$$

Here, $p_0$ is some initial data distribution on $\mathbb{R}^d$, and $W_t$ denotes $d$-dimensional Brownian motion. In the limit $T \to \infty$, the law $p_t$ of the process $X_T$ quickly approaches that of the standard Gaussian measure $\gamma_d$; in particular, we have $p_t \to \gamma_d$ exponentially fast in $t$ in the KL divergence, total variation metric and 2-Wasserstein metric Bakry et al. (2014); Villani (2021). The SDE can be solved explicitly and, at each fixed time $t$, the solution coincides in law with the random variable

$$X_t = e^{-t}X_0 + \sqrt{1 - e^{-2t}}\xi, \ X_0 \sim p_0, \ \xi \sim N(0, I_d).$$

In particular, $X_t$ conditioned on $X_0$ is a Gaussian random variable with mean $e^{-t}X_0$ and variance $1 - e^{-2t}$. By averaging over $X_0$, we obtain a simple expression for the density $p_t$ in terms of $p_0$:

$$p_t(x) = Z_t^{-1} \int \exp\left(-\frac{(x - e^{-t}y)^2}{2(1 - e^{-2t})}\right) dp_0(y) \tag{3}$$

where $Z_t = (2\pi(1 - e^{-2t}))^{d/2}$ is the time-dependent normalization constant.

The reverse process, $\bar{X}_t = X_{T-t}$, is also a diffusion process Anderson (1982); Haussmann and Pardoux (1986), solving the SDE (for $0 \le t \le T$)

$$\bar{X}_t = (\bar{X}_t + 2\nabla \log p_{T-t}(\bar{X}_t))dt + \sqrt{2}d\bar{W}_t, \ \bar{X}_0 \sim p_t, \tag{4}$$

where $\bar{W}_t$ denotes time-reversed Brownian motion on $\mathbb{R}^d$. In order to implement SGMs in practice, one must discretize the OU process, and a canonical algorithm is the exponential integrator scheme (EIS). In order to implement the EIS, one samples $\bar{X}_0^{dis} \sim p_T$, picks time steps $0 = t_0 < t_1 < \cdots < t_N \le T$ and simulates the SDE

$$d\bar{X}_t^{dis} = \left(\bar{X}_t^{dis} + 2\nabla \log p_{T-t_k}(\bar{X}_{t_k})\right) dt + d\bar{B}_t$$

for each interval $[t_k, t_{k+1}]$. S Also, the reverse process is typically initialized at $\bar{X}_0 \sim \gamma_d$ in practice, because $p_T$ is unknown. However, the error accrued by this choice is small, evidenced by the exponential convergence of $p_T$ to $\gamma_d$ as $T \to \infty$. The process one samples is then obtained by replacing the score function at time $T - t_k$ with a score estimate $\mathbf{s}_k$:

$$\begin{cases} dY_t = (Y_t + 2\mathbf{s}_k(Y_{t_k})) \, dt + dB_t, \ t \in [t_k, t_{k+1}] \\ Y_0 \sim N(0, Id). \end{cases} \tag{5}$$

**Loss function:** To learn the score function at time $t$, a natural objective to minimize is the following least-squares risk, namely,

$$\mathbf{s}_t(t, X) \mapsto \mathbb{E}_{X_t \sim p_t}\left[\|s(t, X_t) - \nabla_x \log p_t(X_t)\|^2\right],$$

for a given estimator $\mathbf{s}_t : \mathbb{R}^d \to \mathbb{R}^d$. However, this risk functional is intractable since, in the generative modeling setting, one does not have access to pointwise data of the score function. However, it can be shown Vincent (2011) that for any $\mathbf{s}_t$,

$$\mathbb{E}_{X_t \sim p_t}[\|\mathbf{s}_t(X_t) - \nabla_x \log p_t(X_t)\|^2] = \mathbb{E}_{X_0 \sim p_0}\left[\mathbb{E}_{X_t \sim p_t|X_0}[\|\mathbf{s}_t(t, X_t) - \Psi_t(X_t|X_0)\|^2]\right] + E,$$

where $E$ is a constant independent of $\mathbf{s}_t$. Here, $\Psi_t(X_t|X_0) = -\frac{X_t - e^{-t}X_0}{1 - e^{-2t}}$ denotes the score function of the forward process conditioned on the initial distribution. Note that the integral on the right-hand side can be approximated on the basis of samples of $p_0$, since the trajectories $(X_t|X_0)$ are easy to generate. This motivates our definition of the population risk at time $t$:

$$\mathcal{R}_t^t(\mathbf{s}_t) = \mathbb{E}_{X_0 \sim p_0}\left[\mathbb{E}_{X_t \sim p_t|X_0}[\|s(t, X_t) - \Psi_t(X_t|X_0)\|^2]\right]. \tag{6}$$

If we define the individual loss function at time $t$ by

$$\ell_t^t(\mathbf{s}_t, x) = \mathbb{E}_{X_t|X_0=x}\left[\|\mathbf{s}_t(X_t) - \Psi_t(X_t|X_0)\|^2\right], \tag{7}$$

then the population risk can be written as

$$\mathcal{R}_t^t(\mathbf{s}_t) = \mathbb{E}_{x \sim p_0}[\ell_t^t(\mathbf{s}_t, x)].$$

We also define the empirical risk associated to the i.i.d. samples $\{X_i\}_{i=1}^N$ by

$$\widehat{\mathcal{R}^t}_t^N(\mathbf{s}_t) = \frac{1}{N} \sum_{i=1}^N \ell_t^t(\mathbf{s}_t, X_i). \tag{8}$$

We then to solve the optimization problem

$$\min_{\mathbf{s}_t \in \mathcal{F}} \widehat{\mathcal{R}^t}_t^N(\mathbf{s}_t)$$

where $\mathcal{F}$ is an appropriately defined class of vector fields on $\mathbb{R}^d$ (e.g., a class of neural networks). The use of the risk functional in 6 to learn the score has been termed *denoising score matching*, because the function $\Psi_t(X_t|X_0)$ is the noise added to the sample $X_0$ along the forward process.

## 2.2 NEURAL NETWORKS

A *fully connected feedforward ReLU neural network* is a function of the form

$$x \mapsto W_L \left( W_{L-1} \left( \cdots W_2 \left( W_1 x + b_1 \right)^+ + b_2 \ldots \right)^+ + b_{L-1} \right)^+ + b_L,$$

where $W_L : \mathbb{R}^{d_{L-1}} \times \mathbb{R}^{d_L}$ are matrices, $b_L \in \mathbb{R}^{d_L}$ are vectors, and, when $x$ is a vector, $(x)^+$ is interpreted as a componentwise mapping. The parameters $(W_i)_{i=1}^L$ and $(b_i)_{i=1}^L$ are called the *weights* and *biases* of the neural network respectively. The number of columns of $W_i$ is called the *width* of the $i$th layer. When $L = 2$, a neural network is called *shallow*, and, when they take values in $\mathbb{R}$, such networks admit a representation

$$x \mapsto \sum_{i=1}^m a_i(\langle w_i, x \rangle + b_i)^+,$$

where $x, w_i \in \mathbb{R}^d$ and $a_i, b_i \in \mathbb{R}$. Neural networks have achieved remarkable success in learning complex, high-dimensional functions. In this work, we study their ability to express the score function of the diffusion process defined in Equation 2. In order to control the generalization error incurred by empirical risk minimization, one must introduce a notion of 'complexity' for neural networks, and a natural notion is the *path norm*, defined for real-valued shallow ReLU neural networks by

$$\|\phi\|_{\text{path}} := \sum_{i=1}^m |a_i| \left( \|w_i\|_1 + |b_i| \right), \quad \phi(x) = \sum_{i=1}^m a_i (w_i^T x + b_i)^{(+)}$$

and extended in a natural way to vector valued/deep ReLU neural networks. It has been shown that the collection of $L$-fold compositions of shallow networks of uniformly bounded path seminorm enjoys good generalization properties in terms of Rademacher complexity.

## 3 PROBLEM AND MAIN RESULTS

We outline our main assumptions on the data distribution.

**Assumption 1.** *The data distribution $p_0$ is absolutely continuous with respect to the standard Gaussian distribution. Throughout the paper, we let $p_0$ denote both the probability distribution and the PDF of the data, and we write $f(x) := \|x\|^2/2 + \log p_0(x)$ for the log-relative density of $p_0$ with respect to the standard Gaussian distribution, so that*

$$p_0(x) = \frac{1}{Z} \exp \left( -\frac{\|x\|^2}{2} + f(x) \right).$$

*We further assume that*

1. *There exist positive constants $r_f, \alpha, \beta$ with $\beta \ll 1$ such that $-\alpha\|x\|^2 \leq f(x) \leq \beta\|x\|^2$ whenever $\|x\| \geq r_f$, and $\alpha, \beta$ satisfy $c(\alpha, \beta) := \frac{4(\alpha+\beta)}{(1-\beta)} < 1$;*

2. *$f$ is continuously differentiable;*

3. $\sup_{\|x\| \leq R} \|\nabla f(x)\|$ grows at most polynomially as a function of $R$;

4. the normalization constant $Z$ satisfies $\frac{(2\pi)^{d/2}}{Z} \leq C$, where $C$ is a constant depending at most polynomially on dimension.

Assumption 1 roughly states that the tail of the data distribution should decay almost as quickly as a standard Gaussian. As an illustrating example, consider the Gaussian mixture model $q(x) \propto e^{-\frac{\|x-x_1\|^2}{2\sigma_{\min}^2}} + \frac{\|x-x_2\|^2}{2\sigma_{\max}^2}$. If we write $f(x) = \|x\|^2/2 + \log q(x)$, then for any $\epsilon > 0$, there exists $r_\epsilon > 0$ such that

$$-\left(\frac{1 - \sigma_{\min}^2 + \epsilon}{2\sigma_{\min}^2}\right) \|x\|^2 \leq f(x) \leq \left(\frac{\sigma_{\max}^2 - 1 + \epsilon}{2\sigma_{\max}^2}\right) \|x\|^2,$$

whenever $\|x\| \geq r_\epsilon$. This shows that Assumption 1 applies to Gaussian mixtures, as long as the bandwidths are suitably constrained.

The benefit of expressing the data distribution in terms of the log-relative density is that it leads to a nice explicit calculation of the score function. In particular, Lemma 1 states that the $j^{\text{th}}$ component of the score function of the diffusion process is given by

$$(\nabla_x \log p_t(x))_j = \frac{1}{1 - e^{-2t}} \left(-x_j + e^{-t}\frac{F_t^j(x)}{G_t(x)}\right), \tag{9}$$

where $F_t^j(x) = \int_{\mathbb{R}^d}(e^{-t}x_j + \sqrt{1 - e^{-2t}}u_j)e^{f(e^{-t}x + \sqrt{1-e^{-2t}}u)}\gamma_d(du)$ and $G_t(x) = \int e^{f(e^{-t}x + \sqrt{1-e^{-2t}}u)}\gamma_d(du)$. The linear growth assumption ensures that the tail $p_0$ has similar decay properties to that of a Gaussian, of which we make frequent use. The other assumptions on $\nabla f$ and the normalization constant are stated for convenience.

In order to obtain tractable bounds on the estimation error of SGMs for learning such distributions, it is necessary to impose additional regularity assumptions on the function $f$.

**Assumption 2.** *[Learnability of the log-relative density] For every $\epsilon > 0$ and $R > 0$, there exists an L-layer ReLU neural network $\phi_f^{R,\epsilon}$ which satisfies*

$$\sup_{\|x\| \leq R} |f(x) - \phi_f^{R,\epsilon}(x)| \leq R\epsilon.$$

*We denote by $\eta(\epsilon, R) = \|\phi_f^{R,\epsilon}\|_{path}$ the path norm of the approximating network as a function of $\epsilon$ and $R$.*

We will generally abbreviate $\phi_f^{R,\epsilon}$ to $\phi_f$ in mild abuse of notation. Assumption 2 is weak because any continuous function can be approximated by neural networks to arbitrary precision on a compact set (Cybenko (1989)). However, we are mainly interested in cases where $\phi_f$ generalizes well to unseen data; this corresponds to $\eta(\epsilon, R)$ growing mildly as $\epsilon \to 0$.

## 3.1 GENERAL RESULTS

Our first result shows that, under Assumptions 1 and 2, the score function can be approximated *efficiently* by a neural network, even in high dimensions. Our result helps to understand the massive success of deep learning-based implementations of SGMs used in large-scale applications. We denote by $\mathcal{NN}_{L,K}$ the set of neural networks from $\mathbb{R}^d$ to $\mathbb{R}^d$ with depth $L$ and path norm at most $K$. Recall that for a class of vector fields $\mathcal{F}$, we denote by $\mathcal{F}^{\text{score},t} = \{x \mapsto \frac{1}{1-e^{-2t}}(-x + e^{-t}f(x)) : f \in \mathcal{F}\}$.

**Proposition 1** (Approximation error for score function). *Suppose assumptions 1 ad 2 hold. Then there exists a class of neural networks $\mathcal{NN}$ with low complexity such that*

$$\inf_{\phi \in \mathcal{NN}^{score,t}} \int_{\mathbb{R}^d} \|\phi(x) - \nabla_x \log p_t(x)\|^2 p_t(x)dx$$

$$= O\left(\max\left(\frac{1}{(1 - e^{-2t})^2}(1 + 2\alpha)^{2d}\epsilon^{2(1 - c(\alpha, \beta))}, \frac{1}{(1 - e^{-2t})^3}\epsilon^{1/2}\right)\right),$$

The class of neural networks is defined precisely in the appendix. A high-level idea of the proof is to show that the map from $f$ to the score function $\nabla_x \log p_t$ has low-complexity in a suitable sense. The next result concerns the generalization error of learning the score function via empirical risk minimization.

**Proposition 2.** *[Generalization error for score function] Set $R_\epsilon = \sqrt{d + \frac{1}{1-c(\alpha,\beta)} \log\left(\frac{(1+2\alpha)^d}{\epsilon t^2}\right)}$ and $\tilde{R}_\epsilon = \sqrt{d - \log(t^6\epsilon^4)}$. The empirical risk minimizer $\widehat{s}$ in $\mathcal{NN}^{score,t}$ of the empirical risk $\widehat{\mathcal{R}}^t$ satisfies the population risk bound*

$$\mathcal{R}^t(\widehat{s}) = O(\epsilon^2),$$

*provided the number of training samples is $N \geq N_{\epsilon,t}$, where $N_{\epsilon,t}$ satisfies*

$$N_{\epsilon,t} = \Omega\left( \max\left( 2^{2L_f+10} d^2 t_0^{-6} (1+2\alpha)^{12d} \epsilon^{-4} \eta^4 \left( R_\epsilon, (1+2\alpha)^{2d} \epsilon^{2(1-2c(\alpha,\beta))} \right), \right.\right.$$

$$\left.\left. 2^{2L_f+10} (1+2\alpha)^{12d} t^{-6-72c(\alpha,\beta)} \epsilon^{-4-48c(\alpha,\beta)} \eta^4 (\tilde{R}_\epsilon, t^6\epsilon^4) \right) \right).$$

The next result describes how the learned score can be used to produce efficient samples from $p_0$, with explicit sample complexity rates.

**Proposition 3.** *[Distribution estimation error of SGMs] Let $\widehat{s}$ denote the empirical risk minimizer in $\mathcal{NN}^{score,t}$ of the empirical risk $\widehat{\mathcal{R}}^t$, let $\widehat{p}$ denote the distribution obtained by simulating the reverse process defined in Equation 5 over the time interval $[t_0, T]$ with $\widehat{s_{t_k}}$ in place of the true score for each discretization point, using the exponential integrator discretization scheme outlined in Section 2.1 with maximum step size $\kappa$ and number of steps $M$. Then, with $T = \frac{1}{2}\left(\log(1/d) + 2d\log(1+2\alpha) + 2(1-c(\alpha,\beta))\log(1/\epsilon)\right)$, $M \geq dT\epsilon$, $\kappa \lesssim \frac{1}{M}$, $t_0 \leq M_\beta(f)^{-2}\epsilon^2$, and $N \geq N_{\epsilon,t_0}$ (where $N_{\epsilon,t_0}$ is as defined in Proposition 2), we have*

$$TV(p_0, \widehat{p}) = O(\epsilon)$$

*with probability $> 1 - poly(1/N)$.*

The proof of the above result has three essential ingredients: Prop 2, which controls the score estimation error for any fixed $t$, an application of Theorem 1 in Benton et al. (2023), which bounds the KL divergence between $p_{t_0}$ and $\widehat{p}$ along the exponential integrator scheme in terms of $\kappa$, $M$, $T$ and the score estimation error, and Lemma 10, which proves that the KL divergence between $p_0$ and $p_{t_0}$ can be bounded by $M_\beta(f)$.

### 3.2 EXAMPLES

We now discuss several concrete examples to which our general theory can be applied.

**Infinite-width networks:** Suppose that $p_0 \propto e^{-\frac{\|x\|^2}{2}+f(x)}$, where $f(x)$ is an infinite-width ReLU network of bounded total variation, i.e., $f(x) = \int_{\mathbb{R}^{d+2}} a(w^T x + b)^{(+)} d\mu(a,w,b)$, where $c > 0$ and $\mu$ is a probability measure on $\mathbb{R}^{d+2}$. For such an $f$, the *Barron norm* is defined as

$$\|f\|_{\mathcal{B}} := \inf_\mu \int_{\mathbb{R}^{d+1}} |a|(\|w\|_1 + |b|)\mu(da, dw, db),$$

where the infimum is over all $\mu$ such that the integral representation holds. The space of all such functions is sometimes referred to as the Barron space or variation space associated to the ReLU activation function. The Barron space has been identified as the 'correct' function space associated with approximation theory for shallow ReLU neural networks, since direct and inverse approximation theorems hold. Namely, for any $f$ in the Barron space and any $R > 0$, there exists a shallow ReLU neural network $f_{NN}$ such that $\sup_{\|x\| \leq R} |f(x) - f_{NN}(x)| \leq R\epsilon$, where $f_{NN}$ has $O(\|f\|_{\mathcal{B}}^2 \epsilon^{-2})$ parameters and $\|f_{NN}\|_{path} \lesssim \|f\|_{\mathcal{B}}$. Conversely, any function which can be approximated to accuracy $\epsilon$ by a network with path norm uniformly bounded in $\epsilon$ belongs to the Barron space. A comprehensive study of Barron spaces can be found in Ma et al. (2022).

Under the Barron space assumption on $f$, we can leverage the linear growth and fast approximation rate of Barron spaces to obtain dimension-independent sample complexity rates for SGMs.

**Proposition 4.** *[Distribution estimation under Barron space assumption] Suppose that $p_0(x) \propto e^{-\|x\|^2/2 + f(x)}$, where $f$ belongs to the Barron space. Let $\|f\|_{\mathcal{B}}$ denote the Barron norm and let $c_f = \inf\{c > 0 : |f(x)| \le c\|x\|\} \le \|f\|_{\mathcal{B}}$. Given $\delta \in (0,1)$, let $\epsilon(\delta)$ be small enough that $\frac{8c_f}{\tilde{R}_\epsilon} \le \delta$, where $\tilde{R}_\epsilon = \sqrt{d - \log(M_{1+\delta/4}(f)^{-10}\epsilon^{-10})}$ Then for all $\epsilon \le \epsilon_0$, the distribution $\widehat{p}$ learned by the diffusion model satisfies*

$$TV(\widehat{p}, p_0) = O(\epsilon),$$

*provided the number of samples $N$ satisfies.*

$$N = \Omega\left(2^{2L_f + 10}\left(1 + \frac{2c_f}{\tilde{R}_\epsilon}\right)^{12d} M_{1+\delta/4}^{12 + 144\delta}\epsilon^{-16 - 192\delta}\|f\|_{\mathcal{B}}^4\right).$$

When $\epsilon$ is small, we essentially require $\epsilon^{-10}$ samples to learn $p_0$ to accuracy $\epsilon$ (up to the prefactors) - this is a significant improvement to classical sample complexity bounds in high dimensions, wherein the rate typically tends to zero as $d \to \infty$. We emphasize that the significance of our contribution is that the *rate* of the sample complexity is independent of dimension, and we leave it as an open direction whether all prefactors can be improved to depend only polynomially on $d$.

**Gaussian mixtures:** Distributions that describe real-world data are often highly *multimodal*, and a natural model for such distributions is the Gaussian mixture model; we assume the initial density to be of the form

$$p_0 = \frac{1}{2}\left(\frac{1}{Z_1}\exp\left(-\frac{\|x - x_1\|^2}{2\sigma_{\min}^2}\right) + \frac{1}{Z_2}\exp\left(-\frac{\|x - x_2\|^2}{\sigma_{\max}^2}\right)\right),$$

where $0 < \sigma_{\min}^2 \le \sigma_{\max}^2$ are the bandwidths and $x_1, x_2 \in \mathbb{R}^d$ are the modes of the distribution. The results that follow can easily be adapted to mixtures with more than two components and with arbitrary weights, but we keep the setting as simple as possible to illustrate the results. Due to the growth condition imposed in Assumption 1, our theory cannot be applied to Gaussian mixtures with arbitrarily small bandwidths; this is discussed further in Appendix E. We prove the following distribution estimation result for Gaussian mixtures.

**Proposition 5.** *[Distribution estimation for Gaussian mixture] Given $\epsilon > 0$, set $R_\epsilon = \sqrt{d + \frac{1}{1 - c(\alpha,\beta)}\log\left(\frac{(1+2\alpha)^d}{\epsilon t^2}\right)}$ and $\tilde{R}_\epsilon = \sqrt{d - \log(t^6\epsilon^4)}$. Let $p_0(x) = \frac{1}{2}\left(\frac{1}{Z_1}e^{-\frac{\|x-x_1\|^2}{2\sigma_{\min}^2}} + \frac{1}{Z_2}e^{-\frac{\|x-x_2\|^2}{2\sigma_{\max}^2}}\right)$ be a mixture of two Gaussians, and fix $\delta \ll 1$. Assume that the bandwidths $\sigma_{\min}^2, \sigma_{\max}^2$ satisfy $c(\alpha,\beta) = \frac{4(\alpha+\beta)}{1 - 2\beta} < 1$, where $\alpha$ and $\beta$ are as defined in Assumption 1. Then there exists an $\epsilon_0$ (depending on $\delta$) such that for any $\epsilon \le \epsilon_0$, the distribution $\widehat{p}$ learned by the SGM satisfies*

$$TV(\widehat{p}, p_0) = O\left((1+2\alpha)^d\epsilon^{1-c(\alpha,\beta)}\right),$$

*provided the number of samples $N$ satisfies $N \ge \max(N_{\epsilon,1}, N_{\epsilon,2})$, where*

$$N_{\epsilon,1} = \Omega\left(2^{2L_f + 10}d^2 t_0^{-6}(1+2\alpha)^{12d}\epsilon^{-4} \cdot \sup_{\|x\| \le R_\epsilon} p_0^{-4}(x)\right)$$

*and*

$$N_{\epsilon,2} = \Omega\left(2^{2L_f + 10}(1+2\alpha)^{12d}t^{-6-72c(\alpha,\beta)}\epsilon^{-4-48c(\alpha,\beta)} \cdot \sup_{\|x\| \le \tilde{R}_\epsilon} p_0^{-4}(x)\right).$$

*As an example, if $\sigma_{\min}^2 = \sigma_{\max}^2 = 1$, then we have*

$$TV(\widehat{p}, p_0) = O(\epsilon)$$

*provided the number of samples satisfies*

$$N = \Omega\left(2^{2L_f + 10}(1+\delta)^{12d}e^{d/2}\epsilon^{-24 - 768\delta}\right).$$

The details of the proof are presented in Appendix E. One technical detail is that we need to be able to approximate the log density by a ReLU neural network so that Assumption 2 is satisfied. Unlike in the previous example, the log-likelihood is *not* a Barron function. However, it can be shown that for any $R$, the restriction of the log-likelihood to $B_R$ can be represented by a composition of two Barron functions, and from this it follows that the log-likelihood can be locally approximated by a ReLU network with two hidden layers.

## 4 CONCLUSION

In this paper, we derived distribution estimation bounds for SGMs, applied to a family of sub-Gaussian densities parameterized by Barron functions. The highlight of our main result is that the sample complexity independent of the dimension. An important message of this work is that, for a data distribution of the form in Assumption 1, a low-complexity structure of the log-likelihood $f$ induces a similar low-complexity structure of the score function. In particular, the score function can be approximated in $L^2$ by a neural network without the curse of dimensionality. Some recent works (Oko et al., 2023; Chen et al., 2023d) have derived distribution estimation bounds under assumptions of *low-dimensionality* on the data; we chose to investigate an approximation-theoretic notion of 'low-complexity', and thus our results are a complement to these existing works.

We conclude by mentioning some potential directions for future research. First, we wonder whether similar results could be achieved if we relax some of our assumptions on the data distribution; for instance, we would like to extend the results to the case where the log-likelihood is allowed to decay arbitrarily quickly. Second, it is not clear whether our estimation rate for the class of densities considered is sharp, and thus obtaining lower bounds for sampling from such densities is an interesting open problem. We conjecture that our generalization error bound can be improved using more refined techniques such as local Rademacher complexities. Finally, obtaining training guarantees of score-based generative models remains an important open problem, and another natural direction for future work would be to study the gradient flow/gradient descent dynamics of score matching under similar assumptions on the target distribution.

## 5 ACKNOWLEDGEMENTS

Frank Cole and Yulong Lu thank the support from the National Science Foundation through the award DMS-2343135.

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
