# Appendix

## Table of Contents

## A  PROPERTIES OF THE SCORE FUNCTION AND PROCESS DENSITY

Let us recall the setup of SGMs. We are given samples from a high-dimensional probability distribution $p_0$, and we wish to learn additional samples. We define the forward process $X_t$ as the solution to the SDE

$$\begin{cases} dX_t = -X_t dt + \sqrt{2}dW_t, \ 0 \le t \le T, \\ X_0 \sim p_0, \end{cases}$$

and we note that the marginal distribution $p_t$ of $X_t$ at time $t$ quickly approaches the standard normal distribution $\gamma_d$. The reverse process $\bar{X}_t = X_{T-t}$ happens to satisfy the SDE

$$\begin{cases} d\bar{X}_t = (\bar{X}_t + 2\nabla_x \log p_{T-t})dt + \sqrt{2}dW_t, \ 0 \le t \le T, \\ \bar{X}_0 = X_T, \end{cases} \tag{10}$$

and so to sample the data distribution $p_0$, we run the SDE in equation 10, but with $\bar{X}_0$ as a standard normal and with the score function $\nabla_x \log p_{T-t}$ replaced by an empirical estimator $\hat{\mathbf{s}}(t, x)$. This is made possible by a technique known as score matching (Hyvärinen and Dayan, 2005; Vincent, 2011), which frames the score estimation as a supervised learning problem.

The key assumption of our analysis is that the data distribution $p_0$ is proportional to $e^{-\frac{\|x\|^2}{2} + f(x)}$, where $f$ has 'low-complexity' in the sense of Assumptions 1 and 2. Under this assumption, we can explicitly compute the score function and derive sub-Gaussian bounds on the forward process density.

**Lemma 1.** *Let $\gamma_d(du)$ denote the standard Gaussian probability measure on $\mathbb{R}^d$. Then under Assumption 1, the $j^{th}$ component of the score function of the diffusion process is given by*

$$(\nabla_x \log p_t(x))_j = \frac{1}{1 - e^{-2t}} \left( -x_j + e^{-t} \frac{F_t^j(x)}{G_t(x)} \right), \tag{11}$$

*where $F_t^j(x) = \int_{\mathbb{R}^d} (e^{-t}x_j + \sqrt{1 - e^{-2t}}u_j)e^{f(e^{-t}x + \sqrt{1-e^{-2t}}u)}\gamma_d(du)$ and $G_t(x) = \int e^{f(e^{-t}x + \sqrt{1-e^{-2t}}u)}\gamma_d(du)$.*

*Proof of Lemma 1.* The forward process density is given by

$$p_t(x) = \frac{1}{Z_t} \int e^{-\frac{\|x - e^{-t}y\|^2}{2(1-e^{-2t})}} e^{-\|y\|^2/2 + f(y)} dy,$$

where $Z_t$ is the normalization constant. We therefore have

$$(\nabla_x p_t(x))_j = \frac{1}{Z_t(1-e^{-2t})} \left( -x_j p_t(x) + e^{-t} \int y_j e^{-\frac{\|x - e^{-t}y\|^2}{2(1-e^{-2t})}} e^{-\|y\|^2/2 + f(y)} dy \right)$$

and thus

$$(\nabla_x \log p_t(x))_j = \frac{1}{1-e^{-2t}} \left( -x_j + e^{-t} \frac{\int y_j e^{-\frac{\|x - e^{-t}y\|^2}{2(1-e^{-2t})}} e^{-\|y\|^2/2 + f(y)} dy}{\int e^{-\frac{\|x - e^{-t}y\|^2}{2(1-e^{-2t})}} e^{-\|y\|^2/2 + f(y)} dy} \right).$$

By completing the square, we have

$$e^{-\frac{\|x - e^{-t}y\|^2}{2(1-e^{-2t})}} e^{-\|y\|^2/2} = e^{-\frac{\|y - e^{-t}x\|^2}{2(1-e^{-2t})}} e^{-\|x\|^2/2},$$

and therefore, after cancellation and an appropriate change of variables, we arrive at

$$
\begin{aligned}
(\nabla_x \log p_t(x))_j &= \frac{1}{1-e^{-2t}} \left( -x_j + e^{-t} \frac{\int y_j e^{-\frac{\|y - e^{-t}x\|^2}{2(1-e^{-2t})}} e^{-\|x\|^2/2 + f(y)} dy}{\int e^{-\frac{\|y - e^{-t}x\|^2}{2(1-e^{-2t})}} e^{-\|x\|^2/2 + f(y)} dy} \right) \\
&= \frac{1}{1-e^{-2t}} \left( -x_j + e^{-t} \frac{\int y_j e^{-\frac{\|y - e^{-t}x\|^2}{2(1-e^{-2t})}} e^{f(y)} dy}{\int e^{-\frac{\|y - e^{-t}x\|^2}{2(1-e^{-2t})}} e^{f(y)} dy} \right) \\
&= \frac{1}{1-e^{-2t}} \left( -x_j + e^{-t} \frac{\int (e^{-t}x_j + \sqrt{1-e^{-2t}} u_j) e^{f(e^{-t}x + \sqrt{1-e^{-2t}}u)} \gamma_d(du)}{\int e^{f(e^{-t}x + \sqrt{1-e^{-2t}}u)} \gamma_d(du)} \right) \\
&:= \frac{1}{1-e^{-2t}} \left( -x_j + e^{-t} \frac{F_t^j(x)}{G_t(x)} \right).
\end{aligned}
$$
(12)

$\square$

The following pointwise sub-Gaussian bounds are used throughout this work.

**Proposition 6.** *Under assumption 1, for all $t > 0$, $\|x\| \geq r_f$,*

$$p_t(x) \lesssim \left( 2\pi (1-2\beta)^{-1} \right)^{-d/2} e^{-\frac{(1-2\beta)\|x\|^2}{2}}$$

*Proof.* By completing the square as in Lemma 1, we have

$$p_t(x) = \frac{1}{Z} (2\pi(1-e^{-2t}))^{-d/2} e^{-\|x\|^2/2} \int_{\mathbb{R}^d} e^{-\frac{\|x - e^{-t}y\|^2}{2(1-e^{-2t})}} e^{f(y)} dy.$$

For $\|x\| \leq r_f$, we then use the quadratic growth and a change of variables to bound $p_t$:

$$
\begin{aligned}
p_t(x) &= \frac{1}{Z} e^{-\|x\|^2/2} \int_{\mathbb{R}^d} e^{f(\sqrt{1-e^{-2t}}u + e^{-t}x)} \gamma_d(du) \\
&\leq \frac{1}{Z} e^{-\|x\|^2/2} \int_{\mathbb{R}^d} e^{\beta\|\sqrt{1-e^{-2t}}u + e^{-t}x\|^2} \gamma_d(du) \\
&\leq \frac{1}{Z} e^{-\|x\|^2/2} \int_{\mathbb{R}^d} e^{\beta(\|u\|^2 + \|x\|^2)} \gamma_d(du) \\
&= \frac{(1-2\beta)^{-d/2}}{Z} e^{-\frac{(1-2\beta)\|x\|^2}{2}} \\
&\lesssim \left( 2\pi (1-2\beta)^{-1} \right)^{-d/2} e^{-\frac{(1-2\beta)\|x\|^2}{2}}
\end{aligned}
$$

$\square$

The following lemma control the growth of $F_t^j$ and $G_t$.

**Lemma 2.** *Let $F_t^j$ and $G_t$ be as defined in Lemma 1. Let $R \geq \max(r_f, \sqrt{\frac{1}{\beta} \sup_{\|x\| \leq r_f} |f(x)|})$. Then*

$$\sup_{\|x\| \leq R} |F_t^j(x)| = O\left((1 - 2\beta)^{-d/2} e^{\beta R^2}\right)$$

*and for $\|x\| \leq R$,*

$$(1 + 2\alpha)^{-d/2} e^{-\alpha R^2} \leq G_t(x) \leq (1 - 2\beta)^{-d/2} e^{-\beta R^2}.$$

*Proof.* For the lower bound on $G_t$, for $\|x\| \leq R$, $R$ sufficiently large, we have

$$G_t(x) = \int_{\mathbb{R}^d} e^{f(e^{-t}x + \sqrt{1 - e^{-2t}}u)} \gamma_d(du)$$

$$\geq \int_{\mathbb{R}^d} e^{-\alpha \|e^{-t}x + \sqrt{1 - e^{-2t}}u\|^2} \gamma_d(du)$$

$$\geq \int_{\mathbb{R}^d} e^{-\alpha(\|x\|^2 + \|u\|^2)} \gamma_d(du)$$

$$\geq (1 + 2\alpha)^{-d/2} e^{-\alpha R^2}.$$

The upper bound on $G_t$ is proven similarly. For the upper bound on $F_t^j$, the proof is similar: we have for $\|x\| \geq R$,

$$|F_t^j(x)| = \left| \int_{\mathbb{R}^d} \left(e^{-t}x_j + \sqrt{1 - e^{-2t}}u_j\right) e^{f(e^{-t}x + \sqrt{1 - e^{-2t}}u)} \gamma_d(du) \right|$$

$$\leq \int_{\mathbb{R}^d} (\|x\| + \|u\|) e^{\beta(\|x\|^2 + \|u\|^2)} \gamma_d(du)$$

$$\leq e^{\beta R^2} \left( \int_{\mathbb{R}^d} e^{\frac{C_f \|u\|^2}{2}} \gamma_d(du) + \int_{\mathbb{R}^d} \|u\| e^{\beta \|u\|^2} \gamma_d(du) \right)$$

$$= O\left((1 - 2\beta)^{-d/2} e^{\beta R^2}\right)$$

$\square$

# B  HIGH-LEVEL PROOF SKETCH

Before delving into the details of the proof, we give an overview of the proof technique.

## B.1  PROOF OVERVIEW OF APPROXIMATION ERROR BOUND

Recall that the $j^{\text{th}}$ component of the score function takes the form $(t, x) \mapsto \frac{1}{1 - e^{-2t}} \left(-x_j + e^{-t} \frac{F_t^j(x)}{G_t(x)}\right)$ where $F_t^j(x) = \int (e^{-t}x_j + \sqrt{1 - e^{-2t}}u_j) e^{f(e^{-t}x + \sqrt{1 - e^{-2t}}u)} \gamma_d(du)$ and $G_t(x) = \int e^{f(e^{-t}x + \sqrt{1 - e^{-2t}}u)} \gamma_d(du)$. We break our approximation argument into two main steps.

**Step 1- approximation of $\frac{F_t^j(x)}{G_t^j(x)}$ on the ball** $B_R$: The first step of the proof is to approximate function $\frac{F_t^j(x)}{G_t^j(x)}$ on a bounded domain. For the function $F_t^j$, note that the integrand can be viewed as a composition of three simple functions, namely the function $x \mapsto f(x)$, the one-dimensional exponential map $x \mapsto e^x$, and the two-dimensional product map $(x, y) \mapsto xy$. By assumption, $f$ can be approximated by a shallow neural network $\phi_f$ with low path-norm. It is well known that the latter two maps can be approximated on bounded domains by a shallow neural networks $\phi_{exp}$ and $\phi_{prod}$; see the Appendix for details. We note that the complexity of the neural network needed to approximate the exponential map $x \mapsto e^x$ on the interval $[-C, C]$ grows exponentially with $C$. However, the fast tail decay of the data distribution ensures that we can restrict attention to an interval which is not too large and still achieve good approximation bounds.

In turn, this allows us to approximate the integrand in the definition of $F_t^j$ by a deep neural network $x \mapsto \phi_{prod}(x_j, \phi_{\exp}(\phi_f(x)))$. We then discretize the integral with respect to $\gamma_d(du)$ using a Monte Carlo sampling argument, and the resulting neural network is

$$\Phi_{F,t}^j(x) = \frac{1}{m} \sum_{i=1}^m \phi_{prod}(e^{-t}x_j + \sqrt{1 - e^{-2t}}(u_i)_j, \phi_{\exp}(\phi_f(e^{-t}x + \sqrt{1 - e^{-2t}}u_i))),$$

for Monte Carlo sample points $\{u_i\}_{i=1}^m$. The procedure for $G_t(x)$ works very similarly, since $G_t(x) = \int e^{f(e^{-t}x + \sqrt{1-e^{-2t}}u)} \gamma_d(du)$. This gives us neural networks $\Phi_{F,t}^j$ and $\Phi_{G,t}$ that approximate $F_t^j$ and $G_t$ on the ball $B_R$. By approximating the quotient map $(x, y) \mapsto \frac{x}{y}$ by another shallow network $\phi_{quot}$, we obtain a deep neural network approximation $\Phi_t^j := \phi_{quot}(\Phi_{F,t}^j(x), \Phi_{G,t}(x))$ for $\frac{F_t^j(x)}{G_t^j(x)}$, valid on $B_R$.

Here, it is crucial that our approximation for the individual functions $F_t^j, G_t$ is in the sup norm; this ensures that the neural network approximation to $G_t$ is bounded away from zero (since $G_t$ itself is bounded away from 0), which in turn allows us to control the approximation error of the quotient map $(x, y) \mapsto \frac{x}{y}$.

**Step 2 - approximation on the unbounded domain** $L^2(p_t)$**, for fixed** $t$: The approximation metric we care about is ultimately not the uniform metric on $B_R$, but the $L^2(p_t)$ metric on all of $\mathbb{R}^d$. To deal with the unbounded approximation domain, we bound the tail of the density $p_t$ and use a truncation argument; in particular, $p_t$ is sub-Gaussian by Lemma 6, so that the truncation error depends mildly on the radius of the ball $B_R$ from Step 1. By choosing the optimal $R$, this gives an approximation of $\frac{F_t^j}{G_t}$ in $L^2(p_t)$ for a fixed time, thus completing the second step of the proof.

## B.2 PROOF OVERVIEW OF GENERALIZATION

Recall that our goal is, for each $t$, to bound the population risk $\mathcal{R}^t(\cdot)$ at the minimizer of the empirical risk over a class of neural networks (to be specified in the detailed proof). For technical reasons, we work with the minimizer $\widehat{\mathbf{s}}$ of a truncated version $\widehat{\mathcal{R}}_R^t$ of the empirical risk, where the data is assumed to be uniformly bounded along the forward process. However, we choose the truncation radius $R$ large enough so that the error incurred by this step is marginal. If we define $\mathcal{R}_R^t$ as the corresponding $R$-truncated version of the population risk, then the generalization error can be decomposed as

$$\mathcal{R}^t(\widehat{\mathbf{s}}) = \underbrace{\left(\mathcal{R}^t(\widehat{\mathbf{s}}) - \mathcal{R}_R^t(\widehat{\mathbf{s}})\right)}_{\text{truncation error}} + \underbrace{\left(\mathcal{R}_R^t(\widehat{\mathbf{s}}) - \mathcal{R}_R^t(\mathbf{s}_R^*)\right)}_{\text{generalization error}} + \underbrace{\left(\mathcal{R}_R^t(\mathbf{s}_R^*) - \mathcal{R}^t(\mathbf{s}^*)\right)}_{\leq 0} + \underbrace{\mathcal{R}^t(\mathbf{s}^*)}_{\text{approximation error}} .$$

Here, $\mathbf{s}^*$ is the minimizer of the population risk over the hypothesis class and $\mathbf{s}_R^*$ is the minimizer of the $R$-truncated risk. The first term represents the error we create from working with the truncated risk rather than the true risk, and we bound it using existing large deviation bounds on the OU process from Oko et al. (2023). Term II represents the generalization error of the truncated risk. In order to bound the Rademacher complexities of some relevant function classes, we need the individual loss function $\ell^t$ to have certain properties (such as boundedness and Lipschitz-continuity). Unfortunately, the loss fails to have these properties fail for the loss $\ell$, but they do hold for it's truncated counterpart $\ell_R^t$, and this is our primary motivation for working with the truncated risk rather than the true risk. In turn, it allows us to apply existing generalization results for neural networks with bounded complexity to our setting.

Term 3 is actually non-negative, because $\mathcal{R}_R^t(\cdot) \leq \mathcal{R}^t(\cdot)$ for any $R$, and hence $\min \mathcal{R}_R^t(\cdot) \leq \min \mathcal{R}^t(\cdot)$. Term 4 is the approximation error, and thus it will be of order $\epsilon^2$, provided the complexity of the hypothesis class is scaled as a suitable function of $\epsilon$. The proof concludes by balancing the truncation radius $R$ and sample size $N$ as suitable functions of $\epsilon$.

## B.3 PROOF OVERVIEW OF DISTRIBUTION ESTIMATION

There are several existing works that bound the distribution estimation error of score-based generative models in terms of the generalization error of the score function. We apply results from Benton et al. (2023) which state that if the score generalization error is $O(\epsilon^2)$ then, provided the parameters

of the sampling algorithm are chosen accordingly and the reverse process is stopped at time $T - t_0$, the learned distribution $\widehat{p}$ satisfies $TV(\widehat{p}, p_{t_0}) = O(\epsilon)$, where $p_{t_0}$ is the distribution of the forward process at time $t_0 \ll 1$. All that remains is to bound $TV(p_{t_0}, p_t)$ and choose $t_0$ as an appropriate function of $\epsilon$. For this, Lemma 10 shows that $TV(p_{t_0}, p_0) = O(\sqrt{t_0})$.

## C  PROOFS FOR APPROXIMATION

The following lemma is a general approximation error bound in the sup norm for functions defined by certain integral representations. We use it to approximate the functions $F_t^j$ and $G_t$, since they admit natural integral representations. The proof adapts the proof of Theorem 1 in Klusowski and Barron (2018). Though the proof idea seems well known, we have not found the general version of the result in the literature, so we state it here in case it may be of interest to others.

**Lemma 3.** *Let $g : \mathbb{R}^d \times \mathbb{R}^p \to \mathbb{R}$ be a function such that for all $K, R > 0$,*

1. *$L_{K,R} := \sup_{\|\theta\| \leq K} \|g(\cdot, \theta)\|_{Lip(B_R)} < \infty$,*

2. *$C_{K,R} := \sup_{\|\theta\| \leq K, \|x\| \leq R} |g(x, \theta)| < \infty$.*

*Here, $\| \cdot \|_{Lip(B_R)}$ is the Lipschitz constant of a function when restricted to the ball of radius $R$. Let $f : \mathbb{R}^d \to \mathbb{R}$ be a function of the form*

$$f(x) = \int_{\|\theta\| \leq K} g(x, \theta) \mu(d\theta),$$

*where $\mu$ is a Radon measure on $\{\|\theta\| \leq K\}$. Then, for any $R > 0$, there exist $(a_i, \theta_i)_{i=1}^n$ with $a_i \in \{\pm 1\}$ and $\|\theta_i\| \leq K$ such that*

$$\sup_{\|x\| \leq R} \left| f(x) - \frac{\|\mu\|_{TV}}{m} \sum_{i=1}^m a_i g(x, \theta_i) \right| \leq \frac{24 \|\mu\|_{TV}}{\sqrt{m}} \cdot C_{K,R} \cdot \sqrt{d \log(m L_{K,R} R)},$$

*Proof.* Notice that by normalizing $\mu$ and decomposing it into positive an negative parts, we can write

$$f(x) = \|\mu\|_{TV} \int_{\{\pm 1\} \times \{\|\theta\| \leq K\}} a g(x, \theta) \tilde{\mu}(da, d\theta),$$

where $\tilde{\mu}$ is a probability measure. Let $(a_i, \theta)_{i=1}^m$ be an i.i.d. sample from $\tilde{\mu}^m$, and define $f_m(x; \Theta) = \frac{\|\mu\|_{TV}}{m} \sum_{i=1}^m a_i g(x, \theta_i)$. We can view $f_m(x; \Theta)$ as an empirical process on the parameter space $\Theta = (a_i, \theta_i)_{i=1}^m$ indexed by $x \in B_R$. Let $\widehat{\mu}$ denote the empirical measure associated to the samples, and notice that for any $x, x' \in B_R$, it holds that

$$\|x - x'\|_{L^2(\widehat{\mu})}^2 \leq \frac{1}{m} \sum_{i=1}^m |g(x, \theta_i) - g(x', \theta_i)|^2 \leq L_{K,R}^2 \|x - x'\|^2.$$

This proves that the covering number of $B_R$ under the $L^2(\widehat{\mu})$ norm is $O(L_{K,R}(R/\epsilon)^d)$. By Dudley's Theorem for empirical processes, it holds that

$$\mathbb{E} \sup_{\|x\| \leq R} |f(x) - f_m(x; \Theta)| \leq \frac{24 \|\mu\|_{TV}}{\sqrt{m}} \inf_{0 \leq t \leq \delta/2} \int_t^{\delta/2} \sqrt{\log \mathcal{N}(B_R, \| \cdot \|_{L^2(\widehat{\mu})}, \epsilon)} d\epsilon,$$

where $\delta = \sup_{\|x\| \leq R} \|x\|_{L^2(\widehat{\mu})}$. We can bound $\delta$ by $C_{K,R}$, and by choosing $t = O(1/m)$ in the infimum in the bound from Dudley's theorem, we find that

$$\mathbb{E} \sup_{\|x\| \leq R} |f(x) - f_m(x; \Theta)| \leq \frac{24 \|\mu\|_{TV}}{\sqrt{m}} \inf_{0 \leq t \leq C_{K,R}} \int_t^{C_{K,R}} \sqrt{\log(L_{K,R}) + d \log(R/t)} dt$$

$$= O\left( \frac{\|\mu\|_{TV}}{\sqrt{m}} \cdot C_{K,R} \cdot \sqrt{\log(L_{K,R}) + \log(Rm)} \right)$$

$\square$

The following lemmas prove that the functions $F_t^j$ and $G_t$ can be well-approximated by neural networks. An important feature of the result is that the approximation is done uniformly on the ball of radius $R$, as opposed to in $L^2$. The main idea of the proof is to discretize the Gaussian integrals defining $F_t^j$ and $G_t$ via Lemma 3, and then exploit the compositional structure of the integrands.

**Lemma 4.** *For any $t \in (0, \infty)$ and $1 \leq j \leq d$, let $G_t(x)$ be as defined in Lemma 1. Then for any $\epsilon > 0$ and any $R \geq r_f$, there exists a neural network $\phi_{t,G}(x)$ such that*

$$\sup_{\|x\| \leq R} |\phi_{t,G}(x) - G_t(x)| = O\left(e^{\beta R^2}\left(\epsilon + (1-2\beta)^{-d/2}e^{-\frac{(1-2\beta)R^2}{2}}\right)\right),$$

*In addition, it holds that $\|\phi_{G,t}\|_{path} = O\left(e^{\beta R^2} \cdot \eta(R, \epsilon)\right)$.*

*Proof.* Recall that

$$G_t(x) := \int e^{f(e^{-t}x + \sqrt{1-e^{-2t}}u)}\gamma_d(du).$$

Fix $R > 0$. We break $G_t$ into two parts:

$$G_t(x) = \int e^{f(e^{-t}x + \sqrt{1-e^{-2t}}u)}\gamma_{d,R}(du) + \int_{\|u\| > R} e^{f(e^{-t}x + \sqrt{1-e^{-2t}}u)}\gamma_d(du)$$

$$:= G_{t,R}(x) + \mathcal{E}_{G,R}(x),$$

where $\gamma_{d,R}(du) = \mathbb{I}_{\|u\| \leq R}(u)\gamma_d(du)$. To proceed, we approximate the local part $G_{t,R}(x)$ of $G_t(x)$ and then control the error term $\mathcal{E}_{G,R}(x)$ using the tail decay of the data distribution. We apply Lemma 3 with $g(x, u) = \exp(f(e^{-t}x + \sqrt{1-e^{-2t}}u))$ and $\mu(du) = \gamma_{d,R}(du)$. In this case, we have $\|\mu\|_{TV} \leq 1$,

$$\sup_{\|x\| \leq R, \|u\| \leq R} g(x, u) \leq e^{2\beta R^2} \quad \text{(by the choice of } R\text{)},$$

and

$$\sup_{\|u\| \leq R} \|g(\cdot, u)\|_{Lip(B_R)} \leq \|e^f\|_{Lip(B_{\sqrt{2}R})} \leq e^{2\beta R^2}\sup_{\|x\| \leq R} \|\nabla f(x)\| := e^{2\beta R^2}D_{f,R}.$$

The conclusion of Lemma 3 states that there exist $u_1, \ldots, u_m$ with $\sup_{1 \leq i \leq m} \|u_i\| \leq R$ such that

$$\sup_{\|x\| \leq R} \left|\int_{\mathbb{R}^d} e^{f(e^{-t}x + \sqrt{1-e^{-2t}}u)}\gamma_{d,R}(du) - \frac{\|\gamma_{d,R}\|_{TV}}{m}\sum_{i=1}^m e^{f(e^{-t}x + \sqrt{1-e^{-2t}}u_i)}\right| = O\left(e^{2\beta R^2}\sqrt{\frac{d}{m}}\right)$$

Now, by Lemma 14, there exists a ReLU neural network $\phi_{f,exp}(x)$ such that

$$\sup_{\|x\| \leq R} \left|\phi_{f,exp}(x) - e^{f(x)}\right| \lesssim e^{\beta R^2}\epsilon.$$

and $\phi_{f,exp}$ satisfies $\|\phi_{f,exp}\|_{path} = O()$. We define $\phi_{G,t}(x) := \frac{\|\gamma_{d,R}\|_{TV}}{m}\sum_{i=1}^m \phi_{f,exp}(e^{-t}x + \sqrt{1-e^{-2t}}u_i)$, which satisfies the approximation bound

$$\sup_{\|x\| \leq R} \left|\phi_{G,t}(x) - e^{f(x)}\right| \leq O\left(e^{2\beta R^2}\sqrt{\frac{d}{m}}\right) + \sup_{\|x\| \leq R}\left|\phi_{f,exp}(x) - \frac{\|\gamma_{d,R}\|_{TV}}{m}\sum_{i=1}^m e^{f(e^{-t}x + \sqrt{1-e^{-2t}}u_i)}\right|$$

$$\leq O\left(e^{2\beta R^2}\sqrt{\frac{d}{m}}\right) + \sup_{\|y\| \leq 3R}\left|\phi_{f,exp}(y) - e^{f(y)}\right|$$

$$= O\left(e^{2\beta R^2}\sqrt{\frac{d}{m}} + e^{\beta R^2}\epsilon\right).$$

If we set $m = \Omega(\epsilon^{-2}de^{2\beta R^2})$, then

$$\sup_{\|x\| \leq R} \left|\phi_{G,t}(x) - e^{f(x)}\right| = O(e^{\beta R^2}\epsilon)$$

It remains to bound the error term $\mathcal{E}_{G,R}(x)$:

$$\mathcal{E}_{G,R}(x) = \int_{\|u\|>R} e^{f(e^{-t}x+\sqrt{1-e^{-2t}}u)}\gamma_d(du)$$

$$\leq \int_{\|u\|>R} e^{\beta(\|x\|^2+\|u\|^2)}\gamma_d(du)$$

$$\leq e^{\beta R^2}(1-2\beta)^{-d/2}e^{-\frac{(1-2\beta)R^2}{2}}.$$

We conclude that

$$\sup_{\|x\|\leq R}|\phi_{t,G}(x) - G_t(x)| = O\left(e^{\beta R^2}\left(\epsilon + (1-2\beta)^{-d/2}e^{-\frac{(1-2\beta)R^2}{2}}\right)\right).$$

To see the path norm bounds on $\phi_{G,t}$, we have

$$\|\phi_{G,t}\|_{\text{path}} = \left\|\frac{\|\gamma_{d,R}\|_{TV}}{m}\sum_{i=1}^{m}\phi_{f,exp}(e^{-t}x+\sqrt{1-e^{-2t}}u_i)\right\|_{\text{path}}$$

$$\leq \frac{1}{m}\sum_{i=1}^{m}\|\phi_{f,exp}(e^{-t}x+\sqrt{1-e^{-2t}}u_i)\|_{\text{path}}$$

$$\leq (1+2R)\|\phi_{f,\exp}\|_{\text{path}} \quad \text{(Behavior of path norm under scaling/translation)}$$

$$\leq (1+2R)\cdot e^{\beta R^2}\cdot \eta(R,\epsilon) \quad \text{(path norm of } \phi_{f,\exp})$$

$$= O\left(e^{\beta R^2}\eta(R,\epsilon)\right)$$

This proves the claim. $\qquad\square$

**Lemma 5.** *For any $t \in (0,\infty)$ and $1 \leq j \leq d$, let $F_t^j(x)$ denote be as defined in Lemma 1. Then for any $\epsilon > 0$ and any $R \geq r_f$, there exists a neural network $\phi_{t,F}^j(x)$ such that*

$$\sup_{\|x\|\leq R}\left|\phi_{t,F}^j(x) - F_t^j(x)\right| = O\left(e^{\beta R^2}\left(\epsilon + (1-2\beta)^{-d/2}e^{-\frac{(1-2\beta)R^2}{2}}\right)\right),$$

*In addition, it holds that $\|\phi_{F,t}^j\|_{path} = O\left(e^{3\beta R^2}\cdot\eta(R,\epsilon)\right)$.*

*Proof.* Recall that

$$F_t^j(x) = \int_{\mathbb{R}^d} h^j(e^{-t}x+\sqrt{1-e^{-2t}}u)\gamma_d(du),$$

where $h^j(y) = y_j e^{f(y)}$. The proof is very similar to that of Lemma 4. Define local and global parts $F_{t,R}^j(x)$ and $\mathcal{E}_{F,R}^j(x)$ of $F$ to as in the proof of Lemma 4 (but with $G_t$ replaced by $F_t^j$). Noting that $\sup_{\|u\|,\|x\|\leq R}h^j(e^{-t}x+\sqrt{1-e^{-2t}}u) \leq \sqrt{2}Re^{2\beta R^2}$ and that $h^j$ is $(Re^{2\beta R^2}D_{f,R}+e^{2\beta R^2}) = O(e^{2\beta R^2})$-Lipschitz on $B_{\sqrt{2}R}$, we can apply Lemma 3 to guarantee the existence of $u_1,\ldots,u_m \in B_{\tilde{R}}$ such that

$$\sup_{\|x\|\leq R}\left|F_{t,R}^j(x) - \frac{\|\gamma_{R,d}\|_{TV}}{m}\sum_{i=1}^{m}h^j(e^{-t}x+\sqrt{1-e^{-2t}}u)\right| \lesssim O\left(e^{2\beta R}\sqrt{\frac{d}{m}}\right).$$

Lemma 15 guarantees the existence of a neural network $\Phi_f^j(x)$ such that

$$\sup_{\|x\|\leq R}\left|\Phi_f^j(x) - h^j(x)\right| \lesssim e^{\beta R^2}\epsilon.$$

If we define the ReLU network $\phi_{F,t}^j(x) := \frac{\|\gamma_{R,d}\|_{TV}}{m}\sum_{i=1}^{m}\Phi_f^j(e^{-t}x+\sqrt{1-e^{-2t}}u)$, then an application of the triangle inequality shows that

$$\sup_{\|x\|\leq R}\left|F_{t,R}^j(x) - \phi_{F,t}^j(x)\right| = O\left(\sqrt{\frac{d}{m}}e^{2\beta R^2} + e^{\beta R^2}\epsilon\right) = O(e^{\beta R^2}\epsilon),$$

for $m = \Omega(\epsilon^{-2}de^{2\beta R^2})$. To conclude, we bound the error term $E_{F,R}(x)$ for $\|x\| \geq R$, as in Lemma 4:

$$
\begin{aligned}
E_{F,R}(x) &= \int_{\|u\| > \tilde{R}} (e^{-t}x_j + \sqrt{1 - e^{-2t}}u_j)e^{f(e^{-t}x + \sqrt{1 - e^{-2t}}u)}\gamma_d(du) \\
&\lesssim \int_{\|u\| > R} (e^{-tR + \sqrt{1 - e^{-2t}}|u_j|})e^{\beta(\|x\|^2 + \|u\|^2)} \\
&= O\left(e^{\beta R^2}(1 - 2\beta)^{-d/2}e^{-\frac{(1 - 2\beta)R^2}{2}}\right).
\end{aligned}
$$

This gives the result. The path norm bound follows a similar argument to that in Lemma 4 and uses the path norm bound for the neural network approximant for $x \mapsto x_j e^{f(x)}$, proved in Lemma 15. $\qquad\square$

**Lemma 6.** *Let $t \in [0, \infty)$, and let $F_t^j$ and $G_t$ be defined as in Lemma 1. Let $\epsilon > 0$ be small enough and $R > 0$ large enough. Then there exists a ReLU neural network $\phi_{t,F,G}^j$ such that*

$$
\sup_{\|x\| \leq R} \left| \phi_{t,F,G}^j(x) - \frac{F_t^j(x)}{G_t(x)} \right| = O\left((1 + 2\alpha)^d(1 - 2\beta)^{-d/2}e^{2(\alpha + \beta)R^2}\left(\epsilon + (1 - 2\beta)^{-d/2}e^{-\frac{(1 - 2\beta)R^2}{2}}\right)\right).
$$

*In addition, we have*

$$
\|\phi_{t,F,G}^j\|_{path} = O\left((1 + 2\alpha)^{3d}(1 - 2\beta)^{-3d/2}e^{6(\beta + \alpha)R^2}\eta(R, \epsilon)\right).
$$

*Proof.* Throughout this proof let us denote $F_t^j(x)$ by $F(x)$, $G_t(x)$ by $G(x)$, and $\phi_{t,F,G}^j(x)$ by $\phi_{F,G}(x)$, since none of the estimates will depend on $t$ or $j$. Recall from Lemma 5 and Lemma 4 that there exist ReLU neural networks $\phi_F$ and $\phi_G$ which approximate $F$ and $G$ on the ball of radius $R$ to error

$$
\left(e^{\beta R^2}\left(\epsilon + (1 - 2\beta)^{-d/2}e^{-\frac{(1 - 2\beta)R^2}{2}}\right)\right)
$$

with respect to the uniform norm, and that in addition the networks satisfy $\|\phi_F\|_{path} = O(e^{3\beta R^2}\eta(R, \epsilon))$ and $\|\phi_G\|_{path} = O(e^{\beta R^2}\eta(R, \epsilon))$ The proof proceeds in two steps:

1. Show that $\frac{\phi_F}{\phi_G}$ approximates $\frac{F}{G}$ on the ball of radius $R$;

2. Show that $\frac{\phi_F}{\phi_G}$ can be approximated by a neural network $\phi_{F,G}$ on the ball of radius $R$.

Notice that

$$
\begin{aligned}
\sup_{\|x\| \leq R} \left| \frac{F(x)}{G(x)} - \frac{\phi_F(x)}{\phi_G(x)} \right| &\leq \sup_{\|x\| \leq R} \left| \frac{1}{G(x)} \right| \cdot \sup_{\|x\| \leq R} |F(x) - \phi_F(x)| \\
&+ \sup_{\|x\| \leq R} \left| \frac{\phi_F(x)}{G(x)\phi_G(x)} \right| \cdot \sup_{\|x\| \leq R} |G(x) - \phi_G(x)|.
\end{aligned}
$$

By Lemma 2, we have for $\|x\| \leq R$ that $F$ and $G$ satisfy the bounds

$$
|F(x)| = O\left((1 - 2\beta)^{-d/2}e^{\beta R^2}\right), \quad (1 + 2\alpha)^{-d/2}e^{-\alpha R^2} \leq G(x) \leq (1 - 2\beta)^{-d/2}e^{\beta R^2},
$$

It follows from the fact that $\phi_F$ and $\phi_G$ approximate $F$ and $G$ *uniformly* on $B_R$ (and the choice of $R$ and $\epsilon$) that

$$
\phi_G(x) \geq \frac{1}{2}(1 + 2\alpha)^{-d/2}e^{-\alpha R^2},
$$

and $\phi_F$ is bounded above by $O\left((1-2\beta)^{-d/2}e^{\beta R^2}\right)$. It follows that

$$
\sup_{\|x\|\le R}\left|\frac{F(x)}{G(x)}-\frac{\phi_F(x)}{\phi_G(x)}\right| \lesssim (1+2\alpha)^{d/2}e^{\alpha R^2}\sup_{\|x\|\le R}|F(x)-\phi_F(x)|
$$
$$
+ (1+2\alpha)^d(1-2\beta)^{-d/2}e^{(2\alpha+\beta)R^2}\sup_{\|x\|\le R}|G(x)-\phi_G(x)|
$$
$$
\lesssim (1+2\alpha)^d(1-2\beta)^{-d/2}e^{2(\alpha+\beta)R^2}\left(\epsilon+(1-2\beta)^{-d/2}e^{-\frac{(1-2\beta)R^2}{2}}\right)
$$

and this concludes the first step of the proof. To approximate $\frac{\phi_F}{\phi_G}$ by a neural network, Lemma 13 states that there exists a neural network $\phi_{quot}:\mathbb{R}^2\to\mathbb{R}$ which satisfies

$$
\sup_{x\in[-r,r],y\in[a,b]}\left|\phi_{quot}(x,y)-\frac{x}{y}\right|\lesssim\epsilon,
$$

where the path norm of $\phi_{quot}$ is $O(ba^{-2}M)$, where $M=\max(r^2,b^2a^{-4})$. We apply this lemma with $r=O\left((1-2\beta)^{-d/2}e^{\beta R^2}\right)$, $a=(1+2\alpha)^{-d/2}e^{-\alpha R^2}$ and $b=(1-2\beta)^{-d/2}e^{\beta R^2}$. We conclude that the network $\phi_{F,G}(x):=\phi_{quot}(\phi_F(x),\phi_G(x))$ satisfies

$$
\sup_{\|x\|\le R}\left|\phi_{F,G}(x)-\frac{F(x)}{G(x)}\right| \le \sup_{\|x\|\le R}\left|\phi_{F,G}(x)-\frac{\phi_F(x)}{\phi_G(x)}\right|+\sup_{\|x\|\le R}\left|\frac{\phi_F(x)}{\phi_G(x)}-\frac{F(x)}{G(x)}\right|
$$
$$
\lesssim \epsilon+(1+2\alpha)^d(1-2\beta)^{-d/2}e^{2(\alpha+\beta)R^2}\left(\epsilon+(1-2\beta)^{-d/2}e^{-\frac{(1-2\beta)R^2}{2}}\right)
$$
$$
= O\left((1+2\alpha)^d(1-2\beta)^{-d/2}e^{2(\alpha+\beta)R^2}\left(\epsilon+(1-2\beta)^{-d/2}e^{-\frac{(1-2\beta)R^2}{2}}\right)\right)
$$

To conclude, we note that $\|\phi_{quot}\|_{\text{path}}=O\left((1-2\beta)^{-3d/2}(1+2\alpha)^{3d}e^{(3\beta+6\alpha)R^2}\right)$, and hence by the definition of the path norm,

$$
\|\phi_{F,G}\|_{\text{path}} = O\left((1-2\beta)^{-3d/2}(1+2\alpha)^{3d}e^{(3\beta+6\alpha)R^2}\cdot\max\left(\|\phi_F\|_{\text{path}},\|\phi_G\|_{\text{path}}\right)\right)
$$
$$
= O\left((1+2\alpha)^{3d}(1-2\beta)^{-3d/2}e^{6(\beta+\alpha)R^2}\eta(R,\epsilon)\right).
$$

$\square$

We are now in position to prove Proposition 7, which controls the approximation error for the function $\frac{F_t^j}{G_t}$ in the $L^2(p_t)$ norm by leveraging the tail decay of $p_t$.

**Proposition 7.** *Let $F_t^j$, $G_t$ be defined as in Lemma 1. Let $\epsilon>0$ be small enough. Then there exists a neural network $\phi_{F,G,t}^j(x)$ such that, with $(s_t)_j=\mathbb{I}_{\|x\|\ge R}\frac{1}{1-e^{-2t}}\left(-x_j+e^{-t}\phi_{F,G,t}^j\right)$, we have*

$$
\int_{\mathbb{R}^d}\|s_t-\nabla_x\log p_t(x)\|^2 p_t(x)dx = O\left(\max\left(\frac{1}{(1-e^{-2t})^2}(1+2\alpha)^{2d}\epsilon^{2(1-c(\alpha,\beta))},\frac{1}{(1-e^{-2t})^3}\epsilon^{1/2}\right)\right),
$$

*where $c(\alpha,\beta):=\frac{4(\alpha+\beta)}{(1-2\beta)}$. In addition, we have*

$$
\|\phi_{t,F,G}^j\|_{path} = O\left((1+2\alpha)^{3d}\epsilon^{-3c(\alpha,\beta)}\eta(R_0,\epsilon)\right),
$$

*where $R_0=\Theta(\sqrt{d+\log(\epsilon^{-1})})$.*

*Proof.* Let us again write $\phi_{F,G,t}^j=\phi_{F,G}$, $F_t^j=F$ and $G_t=G$ for ease of notation. Let $\phi_{F,G}$ be the neural network constructed in Lemma 6. Then, for any fixed $R$, it follows from the definition of

$\mathbf{s}_t$ that

$$\int_{\mathbb{R}^d} \|\mathbf{s}_t - \nabla_x \log p_t(x)\|^2 p_t(x) dx \leq \frac{1}{(1-e^{-2t})^2} \sum_{j=1}^d \sup_{x \in B_R} |\phi^j_{F,G,t}(x) - \frac{F^j_t(x)}{G_t(x)}|^2$$
$$+ \frac{1}{(1-e^{-2t})^2} \int_{\|x\| \geq R} \|\nabla_x \log p_t(x)\|^2 p_t(x) dx = I + II.$$

The first term is approximation error on the bounded domain, and we have

$$I = O\left((1+2\alpha)^{2d}(1-2\beta)^{-d} e^{4(\alpha+\beta)R^2} \left(\epsilon^2 + (1-2\beta)^{-d} e^{-(1-2\beta)R^2}\right)\right)$$

by Lemma 6. The second term is truncation error, and we have

$$II \leq \mathbb{E}_{p_t}[\|\nabla_x \log p_t(x)\|^4]^{1/2} \cdot \left(\int_{\|x\| \geq R} p_t(x) dx\right)^{1/2}$$
$$\lesssim \frac{1}{(1-e^{-2t})^2} e^{-\frac{(1-2\beta)R^2}{4}},$$

where the bound $\mathbb{E}_{p_t}[\|\nabla_x \log p_t(x)\|^4]^{1/2} \lesssim \frac{1}{(1-e^{-2t})^2}$ follows from Lemma 21 in Chen et al. (2023a) and the bound for the second factor follows from the tail decay of $p_t$ derived in Lemma 6. To optimize over the cutoff radius $R$, let us choose $R \geq R_0 = \Omega(\sqrt{d + \log(\epsilon^{-1})})$, so that $(1-2\beta)^{-d} e^{-(1-2\beta)R^2} = \epsilon^2$. Then it follows that term $I$ satisfies

$$I = O\left(\frac{1}{(1-e^{-2t})^2}(1+2\alpha)^{2d} \epsilon^{2(1-c(\alpha,\beta))}\right)$$

and

$$II = O\left(\frac{1}{(1-e^{-2t})^3} \epsilon^{1/2}\right).$$

This gives the bound as stated in the proposition. The path norm follows by setting $R = R_0$ in the path norm bound from Lemma 6. □

## D   GENERALIZATION ERROR OF SCORE ESTIMATE

Recall that we are to study the estimation properties of a hypothesis class of the form

$$\mathcal{NN}^t_{score}(L,K) = \{x \mapsto \frac{1}{1-e^{-2t}}\left(-x + e^{-t}\phi_{NN}(x)\right) : \phi_{NN} \in \mathcal{NN}(L,K)\},$$

where $\mathcal{NN}(L,K)$ is the class of $L$-depth ReLU networks from $\mathbb{R}^d$ to $\mathbb{R}^d$ with path norm bounded by $K$. It follows from the definition of the path norm that functions in $\mathcal{NN}(L,K)$ are $K$-Lipschitz continuous. We also restrict attention to functions in $\mathcal{NN}(L,K)$ which satisfy $|\phi(0)| \leq K$. This mild [1] assumption ensures that the ReLU networks we consider satisfy $\|\phi(x)\| \leq 2K$, which we make frequent use. We will later bound the path norm $K$ in terms of the number of training samples for some concrete examples. We defined the individual loss function at time $t$ by

$$\ell^t(\phi, x) = \mathbb{E}_{X_t|X_0=x}\left[\|\phi(t, X_t) - \Psi_t(X_t|X_0)\|^2\right]$$

and the associated population risk

$$\mathcal{R}^t(\phi) = \mathbb{E}_{x \sim p_0}[\ell^t(\phi, x)].$$

For $0 < S < R$, define

$$\ell^t_{R,S}(\phi, x) = \mathbb{I}_{\|x\| \leq S} \mathbb{E}_{X_t|X_0=x}\left[\|\phi(t, X_t) - \Psi_t(X_t|X_0)\|^2 \cdot \mathbb{I}_{W_R}\right]$$

---

[1] The assumption that $\|\phi(0)\| \leq K$ is mild because the network constructed to approximate the score is uniformly close to the score function around the origin. Therefore, provided $K$ is large enough, the approximating network will satisfy $\|\phi(0)\| \leq K$ anyways.

and

$$\ell_S^t(\phi, x) = \mathbb{I}_{\|x\| \leq S} \mathbb{E}_{X_t | X_0 = x} \left[ \|\phi(t, X_t) - \Psi_t(X_t | X_0)\|^2 \right],$$

where $W_R$ is the event $\{\sup_{t_0 \leq t \leq T} \|X_t\| \leq R + \|X_0\|\}$. We also define

$$\mathcal{R}_{R,S}^t(\phi) = \mathbb{E}_{x \sim p_0}[\ell_{R,S}^t(\phi, x)]$$

and $\mathcal{R}_S^t(\phi) = \mathbb{E}_{x \sim p_0}[\ell_S^t(\phi, x)]$. In other words, $\mathcal{R}_{R,S}^t$ is the truncated version of the population risk, where the expectation is restricted to the event that the process begins in the ball of radius $S$ and remains in the ball of radius $R + S$ throughout the relevant time interval. We will use the following large deviation bound on the OU process from Theorem A.1 in Oko et al. (2023).

**Lemma 7.** *Let $X_t$ denote the OU process. Then there is a universal constant $C > 0$ such that for any $0 < S < R$,*

$$\mathbb{P}\left(\|X_t\| \geq R \text{ for some } t \in [t_0, T] \big| \|X_0\| \leq S\right) \leq \frac{T}{t_0} e^{-\frac{(R-S)^2}{2Cd}}.$$

The following regularity bounds on the truncated loss function will be used later to prove a generalization error estimate.

**Lemma 8.** *Let $s_1(t, x)$, $s_2(t, x) \in \mathcal{F}_{score}$ and write $s_i(t, x) = \frac{1}{1 - e^{-2t}}\left(-x + e^{-t}\phi_i(x)\right)$ for $i = 1, 2$ and $\phi_i \in \mathcal{F}_{NN}$. Then for any $0 < S < R$ and any $x \in \mathbb{R}^d$, we have*

$$\left| \ell_{R,S}^t(s_1, x) - \ell_{R,S}^t(s_2, x) \right| = \begin{cases} O(K(R+S)) \left( \frac{e^{-t}}{1 - e^{-2t}} \right)^2 \mathbb{E}_{X_t | X_0 = x} \|\phi_1(x) - \phi_2(x)\|, & x \in B_S \\ 0, & \|x\| > S. \end{cases}$$

*In addition, the truncated loss function is bounded: for any $x \in \mathbb{R}^d$ and any $\phi \in \mathcal{F}_{NN}$, we have*

$$\ell_{R,S}^t(s, x) = O\left( \left( \frac{e^{-t}}{1 - e^{-2t}} \right)^2 K^2 (R+S)^2 \right).$$

*where $s(t, x) = \frac{1}{1 - e^{-2t}}(-x + e^{-t}\phi(x))$.*

*Proof.* Using the definition of $\ell_{R,S}^t$, we have, for any $x \in \mathbb{R}^d$,

$$\left| \ell_{R,S}^t(\mathbf{s}_1, x) - \ell_{R,S}^t(\mathbf{s}_2, x) \right|$$

$$= \left| \mathbb{I}_{B_S}(x) \left( \frac{e^{-t}}{1 - e^{-2t}} \right)^2 \mathbb{E}_{X_t | X_0 = x, W_R}[\|\phi_1(X_t) - X_0\|^2 - \|\phi_2(X_t) - X_0\|^2] \right|$$

$$\leq 2(2K(R+S) + S)\mathbb{I}_{B_S}(x) \left( \frac{e^{-t}}{1 - e^{-2t}} \right)^2 \mathbb{E}_{X_t | X_0 = x, W_R} \|\phi_1(X_t) - \phi_2(X_t)\|$$

Note that for the first inequality, we have used that the map $x \mapsto \|x\|^2$ is $2R$ Lipschitz on $B_R$, and that $\sup_{i=1,2, \, t_0 \leq t \leq T} \|\phi_i(X_t) - X_0\| \leq 2K(R+S) + S$ under the given assumptions. The proof of boundedness follows similarly: for $\phi \in \mathcal{F}_{NN}$ and $\mathbf{s} = \frac{1}{1 - e^{-2t}}(-x + e^{-t}\phi(x))$, we have by the Cauchy-Schwarz inequality that

$$\left| \ell_{R,S}^t(\mathbf{s}, x) \right| = \left| \mathbb{I}_{B_S}(x) \left( \frac{e^{-t}}{1 - e^{-2t}} \right)^2 \mathbb{E}_{X_t | X_0 = x}[\|\phi(X_t) - X_0\|^2 \cdot \mathbb{I}_{W_R}] \right|$$

$$\lesssim \mathbb{I}_{B_S}(x) \left( \frac{e^{-t}}{1 - e^{-2t}} \right)^2 \left( K^2(R+S)^2 + S^2 \right) = O\left( \left( \frac{e^{-t}}{1 - e^{-2t}} \right)^2 K^2(R+S)^2 \right).$$

$\square$

Before proving the main generalization error bound, we need to control the truncation error incurred from using $\mathcal{R}_{R,S}^t$ in place of $\mathcal{R}^t$.

**Proposition 8.** *For any $r_f < S < R$ and any $\mathbf{s} \in \mathcal{NN}^{score,t}(L,K)$, we have*

$$\left|\mathcal{R}_{R,S}^t(\mathbf{s}) - \mathcal{R}^t(\mathbf{s})\right| = O\left(\left(\frac{e^{-t}}{1-e^{-2t}}\right)^2 K^2 \left((T/t_0)^{1/2}e^{-\frac{(R-S)^2}{4Cd}} + e^{-\frac{(1-2\beta)R^2}{2}}\right)\right).$$

*Proof.* Let $\mathbf{s}(x) = \frac{1}{1-e^{-2t}}\left(-x + e^{-t}\phi(x)\right)$ with $\phi \in \mathcal{NN}(L,K)$. We have

$$\mathcal{R}_{R,S}^t(\mathbf{s}) - \mathcal{R}^t(\mathbf{s}) = \left(\mathcal{R}_{R,S}^t(\mathbf{s}) - \mathcal{R}_R^t(\mathbf{s})\right) + \left(\mathcal{R}_R^t(\mathbf{s}) - \mathcal{R}^t(\mathbf{s})\right) = I + II.$$

For the first term, we have

$$
\begin{aligned}
I &= \left(\frac{e^{-t}}{1-e^{-2t}}\right)^2 \mathbb{E}_{x \sim p_0}\left[\mathbb{I}_{B_S}(x) \cdot \mathbb{E}_{X_t|X_0=x}\left[\|X_0 - \phi(X_t)\|^2 \cdot \mathbb{I}_{W_R^c}\right]\right] \\
&\leq \left(\frac{e^{-t}}{1-e^{-2t}}\right)^2 \mathbb{E}_{x \sim p_0}\left[\mathbb{I}_{B_S}(x) \cdot \mathbb{P}(W_R^c)^{1/2} \cdot \mathbb{E}\left[\|X_0 - \phi(X_t)\|^4\right]^{1/2}\right] \\
&\lesssim \left(\frac{e^{-t}}{1-e^{-2t}}\right)^2 \cdot (T/t_0)^{1/2}\, e^{-\frac{(R-S)^2}{4Cd}} \cdot \mathbb{E}_{x \sim p_0}\left[\mathbb{I}_{B_S}(x)\mathbb{E}_{X_t|X_0=x}\left[\|X_0 - \phi(X_t)\|^4\right]^{1/2}\right].
\end{aligned}
$$

To bound the last term, we have

$$
\begin{aligned}
\mathbb{E}_{x \sim p_0}\left[\mathbb{I}_{B_S}(x)\mathbb{E}_{X_t|X_0=x}\left[\|X_0 - \phi(X_t)\|^4\right]^{1/2}\right] &\lesssim \mathbb{E}_{x \sim p_0}\left[\mathbb{I}_{B_S}(x)\mathbb{E}_{X_t|X_0=x}[\|X_0\|^4 + \|\phi(X_t)\|^4]^{1/2}\right] \\
&\leq \left(\mathbb{E}_{x \sim p_0}\left[\mathbb{I}_{B_S}(x)\left(\|x\|^4 + \mathbb{E}_{X_t|X_0=x}[\|\phi(X_t)\|^4]\right)\right]\right)^{1/2} \\
&\leq \left(\mathbb{E}_{x \sim p_0}[\|x\|^4 \cdot \mathbb{I}_{B_S}(x)] + \mathbb{E}_{x \sim p_t}[\|\phi(x)\|^4]\right)^{1/2} \\
&\lesssim \left(S^4 + K^4\mathbb{E}_{x \sim p_t}[\|x\|^4]\right)^{1/2} \\
&= O(K^2).
\end{aligned}
$$

This proves that

$$I = O\left(\left(\frac{e^{-t}}{1-e^{-2t}}\right)^2 \cdot (T/t_0)^{1/2}\, e^{-\frac{(R-S)^2}{4Cd}} K^2\right).$$

For term $II$, we have

$$
\begin{aligned}
II &= \left(\frac{e^{-t}}{1-e^{-2t}}\right)^2 \int_{\|x\| \geq S} \mathbb{E}_{X_t|X_0=x}[\|X_0 - \phi(X_t)\|^2]p_0(x)dx \\
&\lesssim \left(\frac{e^{-t}}{1-e^{-2t}}\right)^2 \left(\int_{\|x\| \geq S} \|x\|^2 p_0(x)dx + \int_{\|x\| \geq S} K^2\|x\|^2 p_t(x)dx\right) \\
&\lesssim \left(\frac{e^{-t}}{1-e^{-2t}}\right)^2 (2\pi(1-2\beta))^{-d/2}\left(\int_{\|x\| \geq S} \|x\|^2 e^{-\frac{(1-2\beta)\|x\|^2}{2}} + K^2\int_{\|x\| \geq S} \|x\|^2 e^{-\frac{(1-2\beta)\|x\|^2}{2}}dx\right) \\
&\lesssim \left(\frac{e^{-t}}{1-e^{-2t}}\right)^2 K^2 e^{-(1-2\beta)\frac{S^2}{2}}.
\end{aligned}
$$

where we have used the uniform-in-time sub-Gaussian upper bound on the process density from Lemma 6. Combining the bounds for terms $I$ and $II$ gives the bound as stated in the lemma. $\square$

The following lemma controls the Rademacher complexity of our hypothesis class.

**Lemma 9.** *For $t > 0$, $R > 0$, let $\mathcal{NN}^{score,t}(L,K)$ be as defined in Section D and let $\mathbb{S} = \{x_1, \dots, x_N\}$ be a collection of points in $\mathbb{R}^d$. Then*

$$
\begin{aligned}
Rad_N\left(\ell_{R,S} \circ \mathcal{NN}^{score,t}(L,K), \mathbb{S}\right) &:= \mathbb{E}_{\epsilon_i \sim Ber(\{\pm 1\})}\left[\sup_{\mathbf{s} \in \mathcal{NN}^{score,t}(L,K)} \frac{1}{N}\sum_{i=1}^N \epsilon_i \cdot (\ell_{R,S} \circ \mathbf{s})(x_i)\right] \\
&\lesssim 2^L dK^2(R+S)^2\left(\frac{e^{-t}}{1-e^{-2t}}\right)^2 \cdot \frac{1}{\sqrt{N}}.
\end{aligned}
$$

*Proof.* We can assume that $\mathbb{S} \subseteq B_S$, because $\ell_{R,S}(x, \mathbf{s}) = 0$ for any $x \notin B_S$ and any $\mathbf{s}$. By Lemma 8 and Lemma 11 (a vector version of the contraction inequality for Rademacher complexity), it holds that

$$\text{Rad}_N \left( \ell_{R,S} \circ \mathcal{NN}^{\text{score},t}(L, K), \mathbb{S} \right) \leq \sqrt{2} dK(R+S) \left( \frac{e^{-t}}{1 - e^{-2t}} \right)^2 \cdot \text{Rad}_N \left( \mathcal{NN}^{\text{score},t}(L, K), \mathbb{S} \right).$$

Then, since $\mathcal{NN}^{\text{score},t}(L, K) = \{ x \mapsto \frac{1}{1-e^{-2t}} \left( -x + e^{-t}\phi(x) \right) : \phi \in \mathcal{NN}(L, K) \}$, it holds by the scaling and translation properties of Rademacher complexity that

$$\text{Rad}_N \left( \mathcal{NN}^{\text{score},t}(L, K), \mathbb{S} \right) \leq \mathbb{E}_{X_t^i | X_0^i = x_i, W_R} \frac{e^{-t}}{1 - e^{-2t}} \cdot \text{Rad}_N \left( \mathcal{NN}(L, K), \mathbb{S}^{\approx} \right),$$

where $\mathbb{S}^t = \{X_t^1, \ldots, X_t^N\}$ are now random variables. It is well-known(e.g., Lemma 3.13 in Wojtowytsch et al. (2020a)) that

$$\text{Rad}_N \left( \mathcal{NN}(L, K), \mathbb{S}^t \right) \leq \max_i \|X_i^t\|_\infty \cdot 2^L K \cdot \sqrt{\frac{2 \log(2d + 2)}{N}}.$$

Note that on the event $W_R$, we have $\max_i \|X_t^i\|_\infty \leq (R+S)$. Putting everything together gives the desired bound. $\qquad\square$

The following lemma bounds the error between $p_0$ and $p_t$, the forward process at time $t$, by bounding the derivative of the the function $t \mapsto KL(p_t \| p_0)$. We emphasize that the estimate is only useful for short times.

**Lemma 10.** *Define* $M_\beta(f) := \int_{\mathbb{R}^d} \left\| \nabla f \left( \frac{x}{1 - 2\beta} \right) \right\|^2 \gamma_d(dx)$. *For any* $t > 0$, *we have* $D_{KL}(p_t \| p_0) \lesssim M_\beta(f) t$.

*Proof.* It suffices to prove that the time derivative of the KL divergence satisfies

$$\partial_t D_{KL}(p_t \| p_0) \lesssim M_\beta(f). \tag{13}$$

To prove inequality 13, we differentiate the relative entropy:

$$\partial_t D_{KL}(p_t \| p_0) = \partial_t \int p_t(x) \log \left( \frac{p_t}{p_0}(x) \right) dx$$
$$= \int \partial_t \left( p_t \right)(x) \log \left( \frac{p_t}{p_0}(x) \right) dx + \int p_t \partial_t \left( \log \left( \frac{p_t}{p_0}(x) \right) \right) dx.$$

But the second term is equal to zero, because

$$\int p_t \partial_t \left( \log \left( \frac{p_t}{p_0} \right) \right) dx = \int p_t \cdot \frac{\partial_t p_t}{p_t} dx$$
$$= \int \partial_t p_t dx$$
$$= \partial_t \left( \int p_t dx \right) = 0,$$

where the last line follows because $p_t$ is a probability density function for all $t$, and hence $\int p_t dx = 1$ for all $t$. Now, recall that $p_t$ satisfies the *Fokker-Planck* equation $\partial_t p_t(x) = \nabla \cdot (x p_t(x)) + \Delta p_t(x)$ for $t > 0$. This means that, with $\gamma_d(x)$ as the standard Gaussian density,

$$\partial_t D_{KL}(p_t \| p_0) = \int \left( \nabla \cdot (p_t(x)x) + \Delta p_t(x) \right) \log \left( \frac{p_t}{p_0}(x) \right) dx$$
$$= \int \nabla \cdot \left( \nabla \log \left( \frac{p_t}{\gamma_d}(x) \right) p_t(x) \right) \log \left( \frac{p_t}{p_0}(x) \right) dx.$$

Integrating by parts, we have

$$\partial_t D_{KL}(p_t \| p_0) = \int \nabla \cdot \left( \nabla \log \left( \frac{p_t}{\gamma_d}(x) \right) p_t(x) \right) \log \left( \frac{p_t}{p_0}(x) \right) dx$$
$$= - \int \nabla \log \left( \frac{p_t}{\gamma_d}(x) \right) \cdot \nabla \log \left( \frac{p_t}{p_0}(x) \right) p_t(x) dx,$$

since the Gaussian tail decay of $p_t$ ensures that the boundary term vanishes. Now, note from the Cauchy-Schwarz inequality that

$$-\int \nabla \log\left(\frac{p_t}{\gamma_d}(x)\right) \cdot \nabla \log\left(\frac{p_t}{p_0}(x)\right) p_t(x) dx$$

$$= -\int \left(\left(\nabla \log\left(\frac{p_t}{p_0}(x)\right) + \nabla \log\left(\frac{p_0}{\gamma_d}(x)\right)\right) \cdot \nabla \log\left(\frac{p_t}{p_0}(x)\right)\right) p_t(x) dx$$

$$\leq -I(p_t \parallel p_0) + \sqrt{I(p_t \parallel p_0)} \cdot \left(\int \left\|\nabla \log\left(\frac{p_0}{\gamma_d}(x)\right)\right\|^2 p_t(x) dx\right)^{1/2},$$

where

$$I(p_t \parallel p_0) := \int \left\|\nabla \log\left(\frac{p_t}{p_0}(x)\right)\right\|^2 p_t(x) dx$$

is the relative Fisher information of $p_t$ with respect to $p_0$. Using the inequality $ax - a^2 \leq \frac{1}{4}x^2$ for $a > 0$, it follows that

$$\partial_t D_{KL}(p_t \parallel p_0) \leq \frac{1}{4} \int \left\|\nabla \log\left(\frac{p_0}{\gamma_d}(x)\right)\right\|^2 p_t(x) dx.$$

But recall that $p_0(x) = \frac{1}{Z} e^{-\|x\|^2/2 + f(x)}$. Therefore

$$\int \left\|\nabla \log\left(\frac{p_0}{\gamma_d}(x)\right)\right\|^2 p_t(x) dx = \int \|\nabla f(x)\|^2 p_t(x) dx$$

$$\lesssim \int \left\|\nabla f\left(\frac{x}{1 - 2\beta}\right)\right\| \gamma_d(dx)$$

$$:= M_\beta(f).$$

This proves the inequality 10 and hence the original claim as well. $\qquad\square$

We are now in position to prove the main generalization bound. The following result is the same content as Proposition 2, but stated more precisely.

**Proposition 9.** *Let $\mathcal{R}^t$ denote the population risk functional at time $t$, let $\widehat{\mathcal{R}}^t$ denote the associated empirical risk functional, and let $\mathcal{R}_R^t = \mathcal{R}_{2R,R}^t$ and $\widehat{\mathcal{R}}_R^t = \widehat{\mathcal{R}}_{2R,R}^t$ denote the truncated risk functionals as defined in Section D. Let $\widehat{s}$ denote the minimizer of $\widehat{\mathcal{R}}_R^t$ over the neural network class $\mathcal{NN}^{score,t}(L, K)$. Then, with $L = L_f + 5$ and $K = O\left((1 + 2\alpha)^{3d} \epsilon^{-3c(\alpha,\beta)} \eta(\sqrt{d + \log(\epsilon^{-1})}, \epsilon)\right)$, we have*

$$\mathcal{R}^t(\widehat{s}) = O((1 + 2\alpha)^{2d} \epsilon^{2(1 - c(\alpha,\beta))}),$$

*with probability $1 - poly(1/N)$, provided the number of training samples is*

$$N = \Omega\left(2^{2L_f + 10} d^2 t_0^{-6} \epsilon^{-4 - 8c(\alpha,\beta)} \eta^4(R_0, \epsilon)\right).$$

We note that, evidenced by Lemma 8, $\widehat{\mathcal{R}}^t = \widehat{\mathcal{R}}_R^t$ with high probability.

*Proof.* Let $s^* = \mathrm{argmin}_{s \in \mathcal{NN}^{score,t}(L,K)} \mathcal{R}^t(s)$ and $s_R^* = \mathrm{argmin}_{s \in \mathcal{NN}^{score,t}(L,K)} \mathcal{R}_R^t(s)$. Then we have

$$\mathcal{R}^t(\widehat{s}) = \left(\mathcal{R}^t(\widehat{s}) - \mathcal{R}_R^t(\widehat{s})\right) + \left(\mathcal{R}_R^t(\widehat{s}) - \mathcal{R}_R^t(s_R^*)\right) + \left(\mathcal{R}_R^t(s_R^*) - \mathcal{R}^t(s^*)\right) + \mathcal{R}^t(s^*) = I + II + III + IV.$$

Term $I$ is the truncation error; by Lemma 8, we have

$$I = O\left(\left(\frac{e^{-t}}{1 - e^{-2t}}\right)^2 K^2 (T/t_0)^{1/2} e^{-\frac{R^2}{4Cd}} + e^{\frac{-(1 - 2\beta)R^2}{2}}\right).$$

Term $II$ is the generalization error for the truncated risk; by Lemmas 9 and 8 (controlling the Rademacher complexity of the relevant function class and uniformly bounding the loss $\ell_R$) and Theorem 26.4 in Shalev-Shwartz and Ben-David (2014), we have

$$II = O\left(\left(\frac{e^{-t}}{1-e^{-2t}}\right)^3 \cdot 2^L dK^2 R^2 \cdot \sqrt{\frac{1}{N}} + \left(\frac{e^{-t}}{1-e^{-2t}}\right)^2 \cdot K^2 R^2 \cdot \sqrt{\frac{2\log(2/\delta)}{N}}\right),$$

with probability at least $1-\delta$. Term $III$ is nonpositive, because $\mathcal{R}_R^t(\cdot) \leq \mathcal{R}^t(\cdot)$ for all $R, t > 0$, and hence $\inf \mathcal{R}_R^t(\cdot) \leq \inf \mathcal{R}^t(\cdot)$. Term $IV$ is the approximation error; by Theorem 7, it is $O\left((1+2\alpha)^{2d}\epsilon^{2(1-c(\alpha,\beta))}\right)$, where we recall $c(\alpha,\beta) = \frac{4(\alpha+\beta)}{1-2\beta}$, provided that $K = O\left((1+2\alpha)^{3d}\epsilon^{-3c(\alpha,\beta)}\eta(R_0,\epsilon)\right)$ for $R_0 = \Theta\left(\sqrt{d+\log(\epsilon^{-1})}\right)$ and $L = L_f + 5$ (recall that $\eta(R,\epsilon)$ is the path norm of the network needed to approximate $f$ to accuracy $R\epsilon$ uniformly over $B_R$).

To balance terms $I$, $II$ and $IV$, let us first choose $R$ large enough that term $I$ is the same order as the approximation error term $IV$. Due to the exponential decay in $R$ of term $I$, this holds for $R$ only logarithmic in all relevant parameters. Let us also take $\delta$ to be polynomial in $1/N$. It then suffices to balance $N$ and $\epsilon$ so that terms $II$ and $IV$ are of the same order, and up to logarithmic factors this amounts to solving

$$\left(\frac{e^{-t}}{1-e^{-2t}}\right)^3 \cdot 2^{L_f+5} dK^2 \cdot \sqrt{\frac{1}{N}} = \Theta\left(\max\left(\frac{1}{(1-e^{-2t})^2}(1+2\alpha)^{2d}\epsilon^{2(1-c(\alpha,\beta))}, \frac{1}{(1-e^{-2t})^3}\epsilon^{1/2}\right)\right).$$

Since $\left(\frac{e^{-t}}{1-e^{-2t}}\right) = O(t^{-1})$ and $K = O\left((1+2\alpha)^{3d}\epsilon^{-3c(\alpha,\beta)}\eta(R_0,\epsilon)\right)$, it therefore holds that if we have

$$N = \Omega\Bigg(\max\Bigg(2^{2L_f+10}d^2 t_0^{-6}(1+2\alpha)^{12d}\epsilon^{-4}\eta^4\left(R_\epsilon, (1+2\alpha)^{2d}\epsilon^{2(1-2c(\alpha,\beta))}\right),$$

$$2^{2L_f+10}(1+2\alpha)^{12d}t^{-6-72c(\alpha,\beta)}\epsilon^{-4-48c(\alpha,\beta)}\eta^4(\tilde{R}_\epsilon, t^6\epsilon^4)\Bigg)\Bigg)$$

samples (where $R_\epsilon = \sqrt{d + \frac{1}{1-c}\log\left(\frac{(1+2\alpha)^d}{\epsilon t^2}\right)}$ and $\tilde{R}_\epsilon = \sqrt{d - \log(t^6\epsilon^4)}$), our score estimation error is $O(\epsilon^2)$. $\qquad\square$

We now employ existing sampling guarantees to prove that the distribution returned by the SGM is close to the true data distribution.

*Proof of Proposition 3.* By Proposition 9 (which controls the score estimation error) and Theorem 1 in Benton et al. (2023) (which controls the sampling error of SGMs in terms of the score estimation error) we have that

$$KL(p_{t_0} \| \widehat{p}) \lesssim (1+2\alpha)^{2d}\epsilon^{2(1-c(\alpha,\beta))} + \kappa^2 dM + \kappa dT + de^{-2T},$$

where $[t_0, T]$ is the time interval of the forward process, $\kappa$ is the maximum step size for the exponential integrator, and $M$ is the number of iterations of the exponential integrator. Choosing $M, \kappa$, and $T$ to scale with $\epsilon$ as described in the statement of Prop 3 yields that each term is of order at most $(1+2\alpha)^{2d}\epsilon^{2(1-c(\alpha,\beta))}$, and hence $TV(p_{t_0}, \widehat{p}) = O((1+2\alpha)^d\epsilon^{1-c(\alpha,\beta)})$. It now remains to bound $TV(p_{t_0}, p_0)$, and Lemma 10 shows that

$$TV(p_{t_0}, p_0) \lesssim M_\beta(f)t_0^{1/2}.$$

It follows that if we scale $t_0$ so that the above expression is $O((1+2\alpha)^{2d}\epsilon^{2(1-c(\alpha,\beta))})$, then we get $TV(\widehat{p}, p_0) \leq TV(\widehat{p}, p_{t_0}) + TV(p_{t_0}, p_0) \lesssim (1+2\alpha)^d\epsilon^{1-c(\alpha,\beta)}$. The number of samples $N$ required to achieve this error was derived in the proof of Proposition 9.

$\qquad\square$

# E   DISTRIBUTION ESTIMATION AND PROOFS FOR CONCRETE EXAMPLES

We give the proofs of the distribution estimation bounds for the general case and for the concrete examples discussed. These proofs are quite short and follow easily from the score estimation results.

*Proof of Theorem 3.* By Proposition 2, we know that with $N \geq N_{\epsilon,t}$ samples, our score estimator $s_t$ is $O(\epsilon)$-close to the true score at time $t$ in $L^2(p_t)$. By Theorem 1 in Benton et al. (2023), the exponential integrator scheme with parameters as defined in Theorem 3 produces a distribution $\widehat{p}$ which satisfies $TV(\widehat{p}, p_{t_0}) = O(\epsilon)$. All that remains then is to bound $TV(p_0, p_{t_0})$. By Lemma 10, we have under Assumption 1 that $TV(p_{t_0}, p_0) \lesssim M_\beta t_0$. It follows that choosing $t_0$ as defined in Theorem 3 balances $TV(\widehat{p}, p_{t_0})$ and $TV(p_{t_0}, p_0)$, so that $TV(\widehat{p}, p_0) = O(\epsilon)$. $\qquad\square$

*Proof of Theorem 4.* We note that a Barron function $f$ grows linearly with a constant $c_f \leq \|f\|_{\mathcal{B}}$, and therefore, for any $R > 0$, we have for all $\|x\| \geq R$ that $|f(x)| \leq c_f \|x\| = c_f \frac{\|x\|}{\|x\|^2} \leq \frac{c_f}{R} \|x\|^2$. This shows that $f$ satisfies that quadratic growth/decay condition of Assumption 1 with constants $\alpha = \beta = \frac{c_f}{R}$ We choose $\tilde{R}_\epsilon = \Omega(\sqrt{d + \log(t^6\epsilon^4)})$ to be the optimal cutoff radius for the approximation error argument. In addition, we know (Ma et al., 2020; Klusowski and Barron, 2018) that for Barron $f$, there exists a shallow ReLU neural network $\phi$ such that

$$\sup_{\|x\| \leq R} |f(x) - \phi(x)| \leq R\epsilon$$

and $\|\phi\|_{\text{path}} \lesssim \|f\|_{\mathcal{B}}$. The result essentially follows from the estimates in Theorem 3 by replacing $\alpha$ and $\beta$ with $\frac{c_f}{\tilde{R}_\epsilon}$ and replacing $\eta(R, \epsilon)$ with $\|f\|_{\mathcal{B}}$; note that in this case $c(\alpha, \beta) \leq \frac{8c_f}{\tilde{R}_\epsilon} \leq \delta$ by assumption. $\qquad\square$

For the Gaussian mixture example, we first need to show that the log-likelihood has the local approximation property, which is the content of the following Lemma. The approximating network has two hidden layers; the inner layer approximates the density and the outer layer approximates $\log(x)$ on the image of the density.

**Lemma 11.** *Suppose that* $p_0(x) = \frac{1}{2} \left( \frac{1}{Z_1} e^{-\frac{\|x-x_1\|^2}{2\sigma_{\min}^2}} + \frac{1}{Z_2} e^{-\frac{\|x-x_2\|^2}{2\sigma_{\max}^2}} \right)$. *Then for any* $R > 0$, *there exists a ReLU network* $f_{NN}$ *with two hidden layers such that*

$$\sup_{\|x\| \leq R} |f(x) - f_{NN}(x)| \leq \epsilon.$$

*Moreover,* $f_{NN}$ *satisfies* $\|f\|_{path} = O\left(d \cdot \sigma_{\min}^{-1} \cdot \max_{\|x\| \leq R} p_0^{-1}(x)\right)$.

*Proof.* By the theory of spectral Barron functions (e.g., Barron (1993) and Klusowski and Barron (2018)) there exists a shallow ReLU network $f_{mix}$ such that $\sup_{\|x\| \leq R} |f_{mix}(x) - p_0(x)| \leq \epsilon$ and $\|f_{mix}\|_{\text{path}} = O(d\sigma_{\min}^{-1})$. We also know from Wojtowytsch et al. (2020b) that $x \mapsto \log(x)$ can be locally approximated to error $\epsilon$ on $[a, b]$ by a network $f_{\log}$ with $\|f_{\log}\|_{\text{path}} = O(a^{-1})$. We set $a = \min_{\|x\| \leq R} p_0(x)$ and $b = \max_{\|x\| \leq R} p_0(x)$. It is then clear that the network $\tilde{f}_{NN} = f_{\log} \circ f_{mix}$ satisfies $\sup_{\|x\| \leq R} |\log(p_0(x)) - \tilde{f}_{NN}(x)| \leq \epsilon$ and $\|\tilde{f}_{NN}\|_{\text{path}} = O\left(d \cdot \sigma_{\min}^{-1} \cdot\right) \max_{\|x\| \leq R} p_0^{-1}(x)$. To conclude, we note that by Lemma 12, the map $x \mapsto \|x\|^2/2$ can be approximated on $\{\|x\| \leq R\}$ by a network $f_{norm}$ to accuracy $\epsilon$, and $f_{norm}$ can be taken to satisfy $\|f_{norm}\|_{\text{path}} = O(dR^2)$. In turn, the network $f_{NN} = \tilde{f}_{NN} + f_{norm}$ satisfies

$$\sup_{\|x\| \leq R} |f(x) - f_{NN}(x)| \leq \epsilon$$

and $\|f_{NN}\|_{\text{path}} = O\left(d \cdot \sigma_{\min}^{-1} \cdot\right) \max_{\|x\| \leq R} p_0^{-1}(x)$. $\qquad\square$

We are now equipped to give a simple proof of the distribution estimation result for Gaussian mixtures.

*Proof Of Proposition 5.* Suppose

$$p_0(x) = \frac{1}{2}\left(\frac{1}{Z_1}e^{-\frac{\|x-x_1\|^2}{2\sigma_{\min}^2}} + \frac{1}{Z_2}e^{-\frac{\|x-x_2\|^2}{2\sigma_{\max}^2}}\right)$$

is a mixture of two Gaussians. Fix some $0 < \delta \ll 1$. Then there exists an $r_0(\delta) > 0$ (depending on $x_1, x_2, \sigma_{\min}, \sigma_{\max}$) such that

$$-\left(\frac{1+\delta}{2\sigma_{\min}^2}\right)\|x\|^2 \le \log p_0(x) \le -\left(\frac{1-\delta}{2\sigma_{\max}^2}\right)\|x\|^2, \quad \forall \|x\| > r_0(\delta) \qquad (14)$$

and therefore we can write $p_0(x) = \frac{1}{Z}e^{-\|x\|^2/2+f(x)}$, where $f(x)$ satisfies

$$-\alpha\|x\|^2 := -\left(\frac{1+\delta-\sigma_{\min}^2}{2\sigma_{\min}^2}\right)\|x\|^2 \le f(x) \le \beta\|x\|^2 := \left(\frac{\sigma_{\max}^2+\delta-1}{2\sigma_{\max}^2}\right)\|x\|^2$$

whenever $\|x\| > r_0$. For $\delta$ small enough, the assumption the $c(\alpha, \beta) < 1$ holds, for instance, whenever $\sigma_{\min}^2 > \frac{2}{3}$ and $\sigma_{\max}^2 \le 1$. If $\epsilon > 0$ is small enough that $r_0(\delta) \le \min\left(R_\epsilon, \tilde{R}_\epsilon\right)$, then the result then follows from Theorem 3 by using the above values of $\alpha$ and $\beta$ and using the complexity measure determined in Lemma 11 in place of $\eta(R, \epsilon)$. In the special case that $\sigma_{\min}^2 = \sigma_{\max}^2 = 1$, we have $\alpha = \beta = \delta/2$ and $c(\alpha, \beta) \le 4\delta$. In addition, in this case we also have $\sup_{x \le \tilde{R}_\epsilon} p_0^{-4}(x) \lesssim e^{d/2}t^{-3}\epsilon^{-2}$. □

# F  BACKGROUND ON NEURAL NETWORKS

## F.1  PATH NORMS

Recall that a shallow ReLU neural network is a function $\phi : \mathbb{R}^d \to \mathbb{R}^k$ whose $j^{\text{th}}$ component is given by

$$(\phi)_j(x) = \sum_{i=1}^m a_{ij}(w_i^T x + b_i)^{(+)}, \quad w_i \in \mathbb{R}^d, \ a_{ij}, b_i \in \mathbb{R}, \ 1 \le j \le k,$$

and a deep ReLU neural network is a composition of shallow ReLU networks. For scalar valued networks, we define the *path norm* by

$$\|\phi\|_{\text{path}} = \inf \sum_{i=1}^m |a_i| \left(\|w_i\|_1 + |b_i|\right),$$

where the infimum is taken over all choices of parameters $(a_i, w_i, b_i)$ such that $\phi(x) = \sum_{i=1}^m a_i(w_i^T x + b_i)^{(+)}$. We extend the path norm to vector-valued shallow networks by

$$\|\phi\|_{\text{path}} = \max_{1 \le j \le k} \|(\phi)_j\|_{\text{path}}, \ \phi : \mathbb{R}^d \to \mathbb{R}^k$$

and to deep networks by

$$\|\phi\|_{\text{path}} = \inf_{\phi_1, \dots \phi_L} \|\phi_1\|_{\text{path}} \cdots \cdots \|\phi_L\|_{\text{path}},$$

where the infimum is over all representations of $\phi$ as a composition of shallow networks. A more thorough study of path norms can be found in Wojtowytsch et al. (2020a).

The path norm captures how large the weights are in an average (i.e., $\ell^1$) sense. Intuitively, a network with a large path norm is not likely to generalize well to unseen data, because its pointwise values depend on large cancellations. In contrast, networks with small path norm provably generalize well, in the sense of Rademacher complexity. The following result due to Wojtowytsch et al. (2020a) makes this precise.

**Proposition 10.** *Let $\mathcal{NN}_{L,K}$ denote the set of L-layer ReLU networks whose path norm is bounded by $K$. Let $S = \{x_1, \dots x_N\}$ denote a set of points in $\mathbb{R}^d$. Then the empirical Rademacher complexity of $\mathcal{NN}_{L,K}$ is bounded by*

$$Rad(\mathcal{NN}_{L,K}, S) := \mathbb{E}_{\epsilon_i \sim Ber(\{\pm 1\})} \sup_{f \in \mathcal{NN}_{L,K}} \frac{1}{N}\sum_{i=1}^N \epsilon_i f(x_i) \le \max_i \|x_i\|_\infty \cdot 2^L \sqrt{\frac{2\log(2d+2)}{N}}.$$

We will need to approximate some simple functions by shallow ReLU neural networks; the next result shows that we can do this efficiently (in the sense of path norms). We emphasize that these results are known (e.g., in Wojtowytsch et al. (2020b)), but we provide the full proofs for the sake of completeness.

**Lemma 12.** *Let $\epsilon > 0$, $R > 0$, and $-\infty < a < b < \infty$.*

1. *There exists a shallow ReLU neural network $\phi_{exp} : \mathbb{R} \to \mathbb{R}$ with $O(\epsilon^{-2}e^{2b})$ neurons such that $\sup_{x \in [a,b]} |\phi_{exp}(x) - e^x| = O(\epsilon)$. In addition, $\phi_{exp}$ satisfies $\|\phi_{exp}\|_{path} = O(e^b)$.*

2. *There exists a shallow ReLU neural network $\phi_{prod} : \mathbb{R}^2 \to \mathbb{R}$ with $O(\epsilon^{-2}R^4)$ neurons such that*
$$\sup_{(x,y) \in [-R,R]^2} |\phi_{prod}(x,y) - xy| = O(\epsilon).$$
   *In addition, $\phi_{prod}$ satisfies $\|\phi_{prod}\|_{path} = O(R^2)$.*

3. *If $a > 0$, then exists a shallow neural $\phi_{inv} : \mathbb{R} \to \mathbb{R}$ with $O(\epsilon^{-2}b^2a^{-4})$ parameters, such that*
$$\sup_{x \in [a,b]} |\phi_{inv}(x) - (1/x)| = O(\epsilon).$$
   *In addition, $\phi_{inv}$ satisfies $\|\phi_{inv}\|_{path} = O(ba^{-2})$.*

*Proof.* For 1), note that for any $x \in [a, b]$, we have

$$e^x + e^a(x - a + 1) = \int_a^x (x - t)e^t dt$$
$$= \int_a^b (x - t)^+ \mu_{exp} dt,$$

where $\mu_{exp}dt = e^t dt$ (note that $\|\mu_{exp}\|_{TV} \leq e^b$. We apply Lemma 3 with the function $g(x, t) = (x - t)^+$. The Lipschitz constant is bounded at 1 (since ReLU is 1-Lipschitz) and the function values of $g$ over $(a, b)$ are bounded at $2b$. We conclude that there exist $t_1, \ldots, t_m \in [a, b]$ such that the function $\phi_{exp}(x) := \frac{\|\mu_{exp}\|_{TV}}{m} \sum_{i=1}^m (x - t_i)^+$ satisfies

$$\sup_{x \in [a,b]} |e^x + e^a(x - a + 1) - \phi_{exp}(x)| \lesssim \frac{2be^b}{\sqrt{m}} \max(b, -a)\sqrt{\log(mb)} = O\left(\frac{e^b}{\sqrt{m}}\right).$$

If we set $m = O(\epsilon^{-2}e^{2b})$, then the approximation error is $O(\epsilon)$ We note that $\phi_{exp}(x) - (e^a(x - a + 1))$ is also a ReLU network, so that (upon renaming $\phi_{exp}$) we have obtained a neural network approximation to $e^x$ on $[a, b]$. Finally, up to an $O(1)$ summand, we have

$$\|\phi_{exp}\|_{path} = \frac{e^b - e^a}{m} \sum_{i=1}^m |t_i| \leq \max(b, -a)(e^b - e^a) = O(e^b).$$

For 2), first observe that we can approximate the one-dimensional map $x \mapsto x^2$ by a shallow ReLU neural network $\phi_{sq}(x)$ on $[-2R, 2R]$ with $O(\epsilon^{-2}R^2)$ neurons. Indeed, for $x \in [-R, R]$, we can write

$$x^2 = \int_0^x 2(x - t)dt = \int_0^{2R} 2(x - t)^+ dt.$$

Using Lemma 3 (in a similar fashion to part 1) above), we conclude the existence of such an approximating $\phi_{sq}(x) = \frac{4R}{m} \sum_{i=1}^m (x - t_i)^{(+)}$. The path seminorm of $\phi_{sq}$ can be bounded by $O(R)$ using the same argument as in part 1). It follows that $xy = \frac{1}{4}((x + y)^2 - (x - y)^2)$ can be approximated by $\phi_{prod}(x, y) := \frac{1}{4}(\phi_{sq}(x + y) - \phi_{sq}(x - y))$ on $[-R, R]^2$. The number of neurons (and path norm constant) of $\phi_{prod}$ is bounded by the number of neurons (and path norm constant) of $\phi_{sq}$, up to a constant multiple.

For 3), the idea is very similar, so we omit some of the details: if $x \in [a, b]$ with $a > 0$, then we have

$$\frac{1}{x} - \frac{2}{a} + \frac{1}{a^2}x = \int_a^b (x - t)^+ \frac{2}{t^3} dt.$$

The conclusion then follows from another application of Lemma 3, noting that the total variation of the parameter measure above is $\int_a^b \frac{2}{t^3} = O(a^{-2})$. $\qquad\square$

As a consequence of Lemma 12, we can approximate the map $(x, y) \mapsto \frac{x}{y}$ by a neural network, provided that the domain of the second coordinate is bounded away from 0.

**Lemma 13.** *Let $\epsilon > 0$, $R > 0$, and $0 < a < b < \infty$. Let $M = \max(R, \frac{b}{a^2})$. Then there exists a ReLU neural network $\phi_{quot}$ with 2 layers and $O(a^{-4}\epsilon^{-4}R^4M^2)$ parameters such that*

$$\sup_{x \in [-R,R], y \in [a,b]} \left| \phi_{quot}(x, y) - \frac{x}{y} \right| = O(\epsilon).$$

*Moreover, we have $\|\phi_{quot}\|_{path} = O(M^2 b a^{-2})$.*

*Proof.* Let $\bar{\epsilon} = (R+1)^{-1}\epsilon$. By Lemma 12, we can find shallow neural networks $\phi$ and $\psi$ satisfying

$$\sup_{y \in [a,b]} \left| \phi(y) - \frac{1}{y} \right| \le \bar{\epsilon}$$

and

$$\sup_{(x,y) \in [-M,M]} |\psi(x, y) - xy| \le \bar{\epsilon}.$$

Let $\phi_{quot}(x, y) = \psi(x, \phi(y))$. Then

$$\sup_{x \in [-R,R], y \in [a,b]} \left| \phi_{quot}(x, y) - \frac{x}{y} \right| \le \sup_{x \in [-R,R], y \in [a,b]} \left| \frac{x}{y} - x\phi(y) \right|$$
$$+ \sup_{x \in [-R,R], y \in [a,b]} |x\phi(y) - \Phi(x, y)|.$$

For the first term, we have

$$\sup_{x \in [-R,R], y \in [a,b]} \left| \frac{x}{y} - x\phi(y) \right| \le R \sup_{y \in [a,b]} \left| \frac{1}{y} - \phi(y) \right| \le R\bar{\epsilon}.$$

For the second term, note that an inspection of the proof of Lemma 12 shows that $\phi$ is $O(a^{-2})$-Lipschitz, so that, up to a constant factor, we have

$$\phi([a, b]) \subseteq \left[ \phi(0) - \frac{b}{a^2}, \phi(0) + \frac{b}{a^2} \right].$$

This guarantees that

$$\sup_{x \in [-R,R], y \in [a,b]} |x\phi(y) - \Phi(x, y)| := \sup_{x \in [-R,R], y \in [a,b]} |x\phi(y) - \psi(x, \phi(y))|$$
$$\le \sup_{x \in [-R,R], y \in [\phi(0) - \frac{b}{a^2}, \phi(0) + \frac{b}{a^2}]} |xz - \psi(x, z)|$$
$$\le \sup_{(x,y) \in [-M,M]^2} |xz - \psi(xz)| \le \bar{\epsilon}.$$

This proves that

$$\sup_{x \in [-R,R], y \in [a,b]} \left| \phi_{quot}(x, y) - \frac{x}{y} \right| \le (R+1)\bar{\epsilon} = \epsilon.$$

To conclude, we have that $\|\phi_{quot}\|_{path} \le \|\psi\|_{path} \cdot \|\phi\|_{path} = O(ba^2M^2)$. $\qquad\square$

**Lemma 14.** *Let $f$ satisfy Assumption 2 . Then for any $R > \max(r_f, \sqrt{\frac{1}{\beta} \sup_{\|x\| \le r_f} |f(x)|})$ and $\epsilon > 0$, there exists a ReLU neural network $\phi_{f,exp}$ with $(L_f + 1)$ layers such that*

$$\sup_{\|x\| \le R} |\phi_{f,exp}(x) - f(x)| \lesssim e^{\beta R^2} \epsilon.$$

*In addition, $\phi_{f,\exp}$ satisfies $\|\phi_{f,exp}\|_{path} = O(e^{\beta R^2} \cdot \eta(R, \epsilon))$.*

*Proof.* By Assumption 2, there exists an $L_f$-layer ReLU neural network $\phi_f$ with

$$\sup_{\|x\| \le R} |f(x) - \phi_f(x)| \le \epsilon.$$

By Lemma 2, there exists a shallow neural network $\phi_{exp} : \mathbb{R} \to \mathbb{R}$ such that

$$\sup_{z \in [-\alpha R^2, \beta R^2]} |\phi_{exp}(x) - e^x| \lesssim \epsilon$$

and $\|\phi_{exp}\|_{path} = O(e^{\beta R^2})$. In turn, the $(L_f + 1)$-layer ReLU network $\phi_{f,exp} = \phi_{exp} \circ \phi_f$ satisfies

$$
\begin{aligned}
\sup_{\|x\| \le R} |\phi_{f,exp}(x) - e^{f(x)}| &\le \sup_{\|x\| \le R} |\phi_{f,exp}(x) - (\phi_{exp} \circ f)(x)| + \sup_{\|x\| \le R} |(\phi_{exp} \circ f)(x) - e^{f(x)}| \\
&\lesssim e^{\beta R^2} \sup_{\|x\| \le R} |\phi_f(x) - f(x)| + \sup_{z \in [-\alpha\|x\|^2, \beta\|x\|^2]} |\phi_{exp}(z) - e^z| \\
&\lesssim e^{\beta R^2} \epsilon.
\end{aligned}
$$

In addition, it follows that

$$\|\phi_{f,exp}\|_{path} \le \|\phi_{exp}\|_{path} \cdot \|\phi_{exp}\|_{path} \lesssim e^{\beta R^2} \cdot \eta(R, \epsilon).$$

$\square$

**Lemma 15.** *Let $f$ satisfy Assumption 2. Then for any $R > \max(r_f, \sqrt{\frac{1}{\beta} \sup_{\|x\| \le r_f} |f(x)|})$ and $\epsilon > 0$, there exists a ReLU neural network $\Phi_f^j(x)$ such that*

$$\sup_{\|x\| \le R} \left| \Phi_f^j(x) - x_j e^{f(x)} \right| \lesssim e^{\beta R^2} \epsilon.$$

*In addition, we have $\|\Phi_f^j\|_{path} = O(e^{3\beta R^2} \cdot \eta(R, \epsilon))$.*

*Proof.* Let $\phi_{f,exp}$ denote the ReLU network constructed in Lemma 14, so that $\sup_{\|x\| \le R} |\phi_{f,exp}(x) - e^{f(x)}| \lesssim e^{\beta R^2} \epsilon$ and $\|\phi_{f,exp}\|_{path} \lesssim e^{\beta R^2} \cdot \eta(R, \epsilon)$. This also implies that $\sup_{\|x\| \le R} |\phi_{f,exp}(x)| \le C e^{\beta R^2}$ for a universal constant $C \ge 1$. By Lemma 12, there exists a shallow ReLU neural network $\phi_{prod} : \mathbb{R}^2 \to \mathbb{R}$ such that

$$\sup_{|y| \le R, |z| \le C e^{\beta R^2}} |\phi_{prod}(y, z) - yz| \lesssim \epsilon$$

and $\|\phi_{prod}\|_{path} = O(e^{2\beta R^2})$. In turn, the $(L_f + 2)$-layer ReLU network $\Phi_f^j(x) = \phi_{prod}(x_j, \phi_{f,exp}(x))$ satisfies $\|\Phi_f^j\|_{path} \lesssim e^{2\beta R^2} \cdot \|\phi_{f,exp}\|_{path} \lesssim e^{3\beta R^2} \cdot \eta(R, \epsilon)$ and

$$
\begin{aligned}
\sup_{\|x\| \le R} |\Phi_f^j(x) - x_j e^{f(x)}| &\le \sup_{\|x\| \le R} |\Phi_f^j(x) - x_j \phi_{f,exp}(x)| + \sup_{\|x\| \le R} |x_j \phi_{f,exp}(x) - x_j e^{f(x)}| \\
&\le \sup_{|y| \le R, |z| \le C e^{\beta R^2}} |\phi_{prod}(y, z) - yz| + R \sup_{\|x\| \le R} |\phi_{f,exp}(x) - e^{f(x)}\| \\
&\lesssim \epsilon + R e^{\beta R^2} \epsilon \lesssim e^{\beta R^2} \epsilon.
\end{aligned}
$$

$\square$

## F.3 Contraction inequality for vector-valued functions

We present the contraction inequality for vector valued functions, which is a slight modification of Theorem 3 in Maurer (2016). The proof of this result can be found in the aforementioned paper.

**Proposition 11.** *Let $\mathcal{F}$ be a separable class of functions from $\mathbb{R}^d$ to $\mathbb{R}^d$, let $\{x_1, \ldots, x_N\} \subset B_S$ and let $\Psi : \mathcal{F} \times \mathbb{R}^d \to \mathbb{R}$ satisfy*

$$\Psi(f,x) - \Psi(f',x) \leq L\mathbb{E}_{X_t|X_0=X_i}\|f(X_t) - f'(X_t)\|, \ \forall f, f' \in \mathcal{F}, \ x \in \mathbb{R}^d.$$

*Then it holds that*

$$\mathbb{E}_{\epsilon_i} \sup_{f \in \mathcal{F}} \sum_{i=1}^{N} \epsilon_i \Psi(f, x_i) \leq \sqrt{2} L \mathbb{E}_{X_t^i|X_0^i=x_i} \mathbb{E}_{\epsilon} \sup_{f \in \mathcal{F}} \sum_{i,k} \epsilon_{ik} f_k(X_t^i),$$

*where $\{\epsilon_{ik}\}_{1 \leq i \leq N, 1 \leq k \leq d}$ are independent Rademacher random variables and $f_k$ denotes the $k$-th component of $f$.*