# OpenReview forum: "Score-based generative models break the curse of dimensionality in learning a family of sub-Gaussian distributions"
_ICLR.cc/2024/Conference — ICLR 2024 poster_

### Official Review · Reviewer_wRd7 · 2023-10-31

**Soundness:** 4 excellent
**Presentation:** 4 excellent
**Contribution:** 4 excellent
**Rating:** 8
**Confidence:** 3

**Summary:**

An exciting result that studies the generalization of the score-based generative modeling under finite sample & shallow neural networks under assumptions of actual sampling density being in the Baron class of functions.

**Strengths:**

1. Rigorous theoretical analysis of population and empirical losses of score-based diffusion models with explicit bounds.
2. The work is well-written and structured.
3. The work answers an intriguing question if there is a class of functions for which score-based diffusion models break the curse of dimensionality, which by itself is not obvious.

**Weaknesses:**

Limited to shallow networks and Baron class (later I don't consider it as something concerning, just some discussion of potential extensions to other classes/other architectures will be appreciated with potential caveats, e.g., in Appendix authors discuss about extending to deep networks -- won't there be any issues with dimension dependency in Lipshitz constants?).

**Questions:**

Assume the function is say, just Lipshitz-smooth. Can we upper bound the Baron constant with a dimensionally dependent constant?
Why asking -- in constants in proofs there are terms that have said Baron constant in the exponent.
So if for densities that are highly non-convex -- we should expect curse of dimensionality to re-appear?

What is connection of densities that are $-||x||^2$+Baron class perturbation to $(\alpha, \beta)-$dissipative distributions? (convex outside of region)? It seems that this is sub-class of such, and if so, can authors elaborate what obstacles there is to generalise results on such class? (that are common in say analysing Langevin Dynamics generalization).

Baron class assumption needed only for universal approximation results about shallow network, right?

---

> ### Author Response · Authors · 2023-11-16
> **Response to reviewer**
>
> We are very grateful to the reviewer for their encouraging feedback and insightful questions. We address your remarks as follows.
>
> - Note that the only exponential dependence in our constants is on the $L^{\infty}$ norm of $f$, not the Barron norm of $f$. The only dependence on the Barron norm of $f$ in our main result (Theorem 2) is **linear**.
> - As we understand it, a measure $p = \frac{1}{Z} e^{-V(x)}$ is considered $(\alpha,\beta)$ dissipative if $\langle \nabla V(x), x \rangle \geq a\|x\|^{\alpha} - b$ for some $a,b \geq 0$, and $\nabla V$ is $\beta$-Holder continuous. In the case that the potential $V$ satisfies $V(x) = \|x\|^2/2 + f(x)$ where $f$ is a spectral Barron function, you are correct that the density is $(\alpha,\beta)$-dissipative: since $$|\nabla f|_{\infty} < \infty$$
>
> for Barron functions, we find that
>     $$\langle \nabla V(x), x \rangle \geq \frac{1}{2}\|x\|^2 - \frac{\|\nabla f\|_{\infty}}{2}$$
>
> and similarly $V$ is Lipschitz with Lipschitz constant at most $(1+\|\nabla f\|_{\infty})$. Hence $\frac{1}{Z}e^{-\|x\|^2/2+f(x)}$ is $(\alpha,\beta)$-dissipative with $\alpha = 2$ and $\beta = 1$. However, our main assumption on the distribution is a low-complexity assumption, which general $(\alpha,\beta)$-dissipative distributions need not satisfy. Those SGMs for learning $(\alpha,\beta)$-dissipative distributions would in general suffer from curse of dimensionality without further low-dimension or low-complexity assumption.
> - Yes, the Barron assumption is needed for the approximation properties of neural networks.
>
> Note that any changes that we have made to the submitted version of the paper are colored in blue. In order to get under the page limit with the changes, we have commented out some less-important remarks, which we plan to add back in the final version.

---

### Official Review · Reviewer_2C1B · 2023-10-31

**Soundness:** 4 excellent
**Presentation:** 4 excellent
**Contribution:** 2 fair
**Rating:** 6
**Confidence:** 3

**Summary:**

This paper introduces a theoretical setting in which a deep network which minimizes the empirical score matching loss produces approximate samples from a high-dimensional data distribution. The curse of dimensionality is broken by assuming that the data distribution is a small perturbation from the Gaussian distribution. The asymptotic sample complexity exponent is independent of the dimension, with a constant that depends polynomially on the dimension and the Barron norm of the perturbation (but exponentially in its $L^\infty$ norm).

**Strengths:**

While sampling guarantees of diffusion models are now rather well-understood, learning guarantees are much less explored, especially in high dimensions. This paper thus studies a central topic, and is the first (to the best of my knowledge) to obtain results on a truly high-dimensional distribution. The paper is also very clearly written and is enjoyable to read: I commend the authors for managing to give intuition about the several steps of the proof despite its technical difficulty.

**Weaknesses:**

The results have a very big limitation, which is the exponential dependence of the error bound on the $L^\infty$ norm of the perturbation $f$. To avoid the curse of dimensionality, this norm thus needs to be $O(1)$ in the dimension. However, in this setting, diffusion models are unnecessary, as much simpler classical techniques are sufficient. Indeed, the Holley-Stroock perturbation lemma [1] implies that the distribution $p(x)$ satisfies a logarithmic Sobolev inequality with a constant that is $\Omega(1)$ in the dimension. The score matching loss at time $t=0$ then controls the error in KL divergence [2], and knowing the score at time $t=0$ is sufficient to obtain sampling guarantees with Langevin/MALA algorithms [3,4]. Therefore, it is not necessary to approximate the scores at all times with a neural network. After a quick assessment of the proof technique, it seems hard to remove this exponential dependence in this norm as the variance of the Monte-Carlo approximation of the integrals depends exponentially on the $L^\infty$ norm of $f$.

I still think the paper should be accepted, because the paper is the first (to the best of my knowledge) to study the sampling complexity of score approximation in high dimensions, and the proof technique might be useful for further research which extends the results in a more interesting setting. But the authors should acknowledge and properly discuss the limitation mentioned above.

[1] Holley, R., and Stroock, D. Logarithmic Sobolev inequalities and stochastic Ising models. J. Stat. Phys. 46, 5–6 (1987), 1159–1194.

[2] Koehler, Frederic, Alexander Heckett, and Andrej Risteski. “Statistical Efficiency of Score Matching: The View from Isoperimetry.” arXiv, October 3, 2022. http://arxiv.org/abs/2210.00726.

[3] Chewi, Sinho. Log-Concave Sampling. https://chewisinho.github.io/main.pdf.

[4] Altschuler, Jason M., and Sinho Chewi. “Faster High-Accuracy Log-Concave Sampling via Algorithmic Warm Starts.” arXiv, February 20, 2023. http://arxiv.org/abs/2302.10249.

**Questions:**

I have some minor remarks and questions:
- Since $f$ is defined up to a constant, the $L^\infty$ norm of $f$ can be replaced with the "oscillation" of $f$ $(\sup f - \inf f)/2$, which makes the invariance of $p(x)$ on addition of constants to $f$ explicit
- The specification of the function class for neural networks makes no mention of the intermediate widths. In general, I was initially confused whether they were assumed to be one or not, as the previous paragraph talks about real-valued shallow networks.
- The equation (9) seems related to the Miyasawa/Tweedie relationship for the structure of the score (see e.g. equation (4) in [5]), as is commonly used in denoising score matching [6]. I think it would make the paper clearer to state that the network has to approximate the conditional expectation $\mathbb E[x_0 | x_t]$, which is essentially the ratio of the functions $F_t$ and $G_t$.
- In section 4.2, it would help the reader to state that the approximations of the intermediate functions (product, exponential, etc) are in the $L^\infty$ norm restricted to balls.

Typos:
- Lemma 4, "K0"
- Lemma 12, extra $|\ell_{R,S}$

[5] Raphan, Martin, and Eero P. Simoncelli. “Least Squares Estimation Without Priors or Supervision.” Neural Computation 23, no. 2 (February 2011): 374–420.

[6] Vincent, Pascal. “A Connection Between Score Matching and Denoising Autoencoders.” Neural Computation 23, no. 7 (July 2011): 1661–74.

---

> ### Author Response · Authors · 2023-11-16
> **Response to reviewer**
>
> We thank the reviewer for their detailed and insightful comments. We would like to address them as follow.
>
> - We do recognize that the $L^{\infty}$ norm of $f$ may depend on the dimension in many contexts. However, as we mentioned in our response to reviewer 1, such exponential dependence on dimension in the prefactors is generally unavoidable.
> - It is a sharp observation that, since the density we study satisfies LSI, it would be enough to learn the score at time 0 and run a Langevin process with the empirical score. To show that distributions which do not satisfy functional inequalities can be learned by SGMs without the curse of dimensionality is therefore an interesting open problem. However, we feel that our bounds are still meaningful because, in practice, one cannot use such structural information about the data to design the generative model, as it is unknown to the learner.
> - About intermediate widths, the proof of Theorem 3 shows that for approximation, it is enough to assume the intermediate widths are all 1, since we're really just approximating a bunch of single-variable functions. However, this is obviously not done in practice, and it does not hurt our generalization argument to assume that intermediate widths are arbitrarily large, as long as the path norm at each layer is bounded independently of the width. We have added a statement about this in the paper (in order to keep under the page limit, we have made this comment in the appendix, specifically at the end of Appendix C).
> - We have added a remark after the statement of Theorem 2 to mention that, without loss of generality, the $L^{\infty}$ norm of $f$ can be replaced with oscillation of $f$, i.e., that the $L^{\infty}$ norm of $f$ can be assumed minimal over all translates. Thank you very much for pointing this out to us.
> - We have stated in Section 3 that the network is estimating the conditional expectation $\mathbb{E}[x_0|x_t]$, which is basically just $F_t/G_t$ under our assumptions. We have also made clearer in Section 4.2 that the intermediate function approximation is only done locally, i.e., that the metric is the $L^{\infty}$ norm restricted to a ball.
> - Thank you for pointing out the typos; we have corrected them.
> - We have added the references you listed to our bibliography.
>
> Note that any changes that we have made to the submitted version of the paper are colored in blue. In order to get under the page limit with the changes, we have commented out some less-important remarks, which we plan to add back in the final version.

---

> > ### Comment · Reviewer_2C1B · 2023-11-20
> >
> > I thank the authors for their response. **I however insist that the authors should acknowledge explicitly in the text that in their setup the target distribution satisfies an LSI.** As of now, I think the paper is misleading regarding the significance of its contributions. It shows that diffusion models beat the curse of dimensionality for a family of distributions **for which classical algorithms already beat the curse of dimensionality.** I agree with the authors that it remains a valuable contribution nonetheless, but this need to be stated clearly for the reader to be able to understand the contributions of the paper.
> >
> > Additionally, there is a typo in the added text, which reads $\mathbb E[x_t | x_0]$ instead of $\mathbb E[x_0 | x_t]$.

---

> > > ### Author Response · Authors · 2023-11-20
> > > **Response to reviewer**
> > >
> > > We thank the reviewer for further comments on the paper.
> > >
> > > We agree with the reviewer that, if the target distribution satisfies a LSI (with a constant which may be dimension-free), then Langevin dynamics with the true score converge exponentially fast, with a rate depending only on the LSI constant. However, in the generative modeling set-up, there is additional error incurred from estimating the score function, and under the LSI alone, this source of error may suffer from the curse of dimensionality. A consequence of our result is that this estimation error has a dimension-independent rate with respect to the number of samples when the density with respect to the Gaussian measure has low-complexity structure.
> > >
> > > We have added a paragraph after Assumption 1 to explain this and provide further context for our results.
> > >
> > > Also, the typo in the added text is fixed. Thank you for pointing that out.

---

> > > > ### Comment · Reviewer_2C1B · 2023-11-23
> > > >
> > > > Thank you for the added paragraph.
> > > >
> > > > Indeed, it is important to consider the approximation of the score function. But again, under the LSI one only needs to estimate the score at time $t=0$, which is here trivial since the assumption is that the log-density has low complexity (unless I'm missing something subtle about the gradients of Barron functions). So I maintain that in the setting of the paper, which implies LSI + low-complexity density, classical algorithms have end-to-end provable theoretical guarantees for generative modeling (both learning and sampling) without needing most of the theoretical machinery of the paper.

---

### Official Review · Reviewer_r8Sf · 2023-11-02

**Soundness:** 3 good
**Presentation:** 4 excellent
**Contribution:** 3 good
**Rating:** 8
**Confidence:** 3

**Summary:**

Recent progress in theoretically understanding score-based diffusion models has shown that sampling (i.e., training a generative model) often reduces to L^2 (or L^\infty) score estimation [Lee et al., 2023; Chen et al., 2023; Benton et al. 2023]. In light of these results, the logical next step is to understand when neural networks can efficiently learn score functions.

This work represents a significant step in this direction. The authors restrict their attention to distributions that can be represented as exponentially tilted Gaussian distributions. Then, they show that if the tilting function, which characterizes the target distribution, is of “low-complexity”, then neural networks can *efficiently* approximate the corresponding score function at a rate that is independent of the data dimension. Building on this approximation-theoretic result for the score function, the paper then establishes a sample complexity upper bound for approximating the target distribution in TV distance. Notably, this upper bound avoids the cursed exponential-in-d rate.

These results are distinct from other recent results which achieve dimension-independent rates by assuming that the target distribution is supported on a low-dimensional manifold.

**References**
- Holden Lee, Jianfeng Lu, and Yixin Tan. Convergence for score-based generative modeling with polynomial complexity. *NeurIPS* 2022.
- Sitan Chen, Sinho Chewi, Jerry Li, Yuanzhi Li, Adil Salim, and Anru R Zhang. Score approximation, estimation and distribution recovery of diffusion models on low-dimensional data. *ICLR,* 2023.
- Joe Benton, Valentin De Bortoli, Arnaud Doucet, and George Deligiannidis. Linear convergence bounds for diffusion models via stochastic localization. *arXiv:2308.03686*, 2023.

**Strengths:**

This is a very well-executed paper. The question addressed in this work is a natural continuation of the recent research effort devoted to understanding theoretical aspects of score-based diffusion models. The main results are timely and significant, and the authors draw ideas (tilted Gaussians, Barron space, path norms) from diverse research areas, and combine them in an elegant manner.

Moreover, the paper is exceptionally well-written, especially considering the technicality of the topic. Ideas are presented clearly and concisely, and intuition behind the proofs are communicated effectively.

**Weaknesses:**

I only have minor editorial comments.

- It would help motivate the setting better if examples of non-trivial target distributions that satisfy the assumptions were provided. What are some well-known distributions which are Gaussians exponentially tilted by a “low-complexity” function from Barron space?
- The sudden appearance of L=8 in Theorem 2 is unnecessarily surprising and mysterious. It would be helpful if this was explained prior to Theorem 2. For example, one can mention that the score function is a composition of the tilting function, which can be approximated by a shallow network, and basic function/arithmetic primitives like exponentiation and multiplication, which can be implemented by stacking more layers.
- Typo in p.9: “allowed to be grow”
- I would suggest simply adding a table of contents to the Appendix (there is a latex package for this) instead of Section B.

**Questions:**

- Aside from the Barron space assumption, what conditions must the tilting function satisfy to ensure that corresponding function is a probability density? An immediate one seems to be subquadratic growth, but are there other conditions that one needs to or would want to assume?
- What are some well-known distributions which are Gaussians exponentially tilted by a “low-complexity” function from Barron space?

---

> ### Author Response · Authors · 2023-11-16
> **Response to reviewer**
>
> We are very grateful for the positive feedback and helpful suggestions from the reviewer. We have addressed the remarks below:
> -  We would first like to mention that, upon further inspection of the proofs, the approximation and generalization proofs can be adapted to the case that the function $f$ is allowed to grow linearly at infinity, say, if $|f(x)| \leq C_f \|x\|$ whenever $\|x\|$ is sufficiently large. Under this change, the main results can be extended to a larger class of distributions, where the function $f$ need not be globally bounded, at the expense of a slightly slower estimation rate. We will update the changes in the final revised version of the paper accordingly.
>
> A motivating example that falls into the linear growth framework for $f$ is the case where the initial density is a Gaussian mixture. Suppose that
>     $$ p(x) = \frac{1}{Z} \left( e^{-\|x-x_1\|^2/2} + e^{-\|x-x_2\|^2/2} \right), x_1, x_2 \in \mathbb{R}^d,.
>     $$
>     Then the log-relative density with respect to the standard normal distribution is
>     $$f(x) = \log(1/Z) + \log \left( e^{-\|x-x_1\|^2/2} + e^{-\|x-x_2\|^2/2} \right) + \frac{\|x\|^2}{2}.$$
> - The depth of the network constructed in Theorem 2 actually should be $L = 7$, and it is explained further in the proof of Theorem 3 in the paper; in essence, it arises from the compositional structure of the score function under our assumption, and the fact that every elementary function that we approximate (e.g., the exponential or product function) corresponds to a single layer in the final network. We are glad that you made us aware that this sudden introduction of depth appears unnatural to the reader, so we have added a brief explanation of where it comes from after the statement of the theorem.
> - Thank you for pointing out the typo, and for suggesting the inclusion of a table of contents for the appendix in place of an extra section. We have edited the paper accordingly.
> - As you point out, for $\frac{1}{Z}e^{-\|x\|^2/2 + f(x)}$ to be a density, we really just need $\|x\|^2/2 - f(x)$ to grow fast enough that $e^{-\|x\|^2/2 + f(x)}$ is integrable. The Barron assumption on $f$ immediately implies that $f$ is bounded hence leading to the function $e^{-|x|^2/2+f}$ being integrable.
>
> Note that any changes that we have made to the submitted version of the paper are colored in blue. In order to get under the page limit with the changes, we have commented out some less-important remarks, which we plan to add back in the final version.

---

### Official Review · Reviewer_bdyJ · 2023-11-06

**Soundness:** 3 good
**Presentation:** 3 good
**Contribution:** 2 fair
**Rating:** 5
**Confidence:** 3

**Summary:**

The authors prove a bound on the sampling error of score based modeling under the assumption that the target density has the form $p_0(x) \\propto e^{f(x)} \\gamma_d(dx)$, where $f$ has $\\|f\\|_\\infty < \\infty$ and $\\|f\\|\_{\\mathcal{B}} < \\infty$. The latter norm is the spectral Barron norm, which is known to define a function class $\\mathcal{B} = \\{f : \\mathbb{R}^d \\to \\mathbb{R}, \\|f\\|\_{\\mathcal{B}} < \\infty\\}$ that can be well approximated in sup-norm by shallow neural networks with bounded path norm. Under these assumptions, the authors bound on the total variation $\\text{TV}(p_0, \\hat{p}_{t_0})$, where $\\hat{p}_{t_0}$ is the output of an early-stopped SGM. The sample complexity of this bound is $\\text{TV}(p_0, \\hat{p}_{t_0}) \\lesssim N^{-\\alpha}$ where $\\alpha > 0$ is independent of the dimension.

 It has been shown in previous works that the SGM sampling error is controlled by the approximation error of the learned score model, $\\hat{s}(t, X) \\approx \\nabla_x \\log p_t(x)$, where $(p_t)_{t \\geq 0}$ is the law of an Ornstein-Uhlenbeck Process with initial condition $p_0$ given by the data distribution. Using integration by parts, the score can be rewritten as $$
    \\nabla_x \\log p_t(x)  = \\frac{1}{1-e^{-2t}}\\left( -x + e^{-t} \\mathbb{E}[ e^{-t} x + \\sqrt{1-e^{-2t}} U \\mid X_t = x] \\right)$$
 where the expectation is taken with respect to $\\text{Law}(U \\mid X_t = x) \\propto e^{f(e^{-t}x + \\sqrt{1-e^{-2t}}U)} \\gamma_d(dU)$. The key insight of the paper is that the conditional expectation in this expression inherits the smoothness properties of $p_0$, and in particular, it can be approximated by a composition of shallow neural networks. In Theorem 3, the authors prove an approximation error bound on the score network by an appropriate class of neural networks. In Theorem 2, using standard techniques from empirical process theory, the authors use this bound to prove the TV bound on SGM sampling error.

**Strengths:**

- Clarity: while there are some minor typos, the results in this paper are clear and well-explained. They are correct to the best of my knowledge.
- Dimension free sample complexity: it is interesting and potentially valuable to prove generalization bounds that depend on instance-specific complexity measures, like the Barron norm of $f$, in place of a dependence on the ambient dimension.

**Weaknesses:**

- Exponential dependence on $\\|f\\|\_{\\infty}$. The assumption that $\\|f\\|\_\\infty < \\infty$ seems extremely strong and the constants in this bound depend exponentially on $\\|f\\|\_{\\infty}$. As a thought experiment, suppose $p_0(x) = \\prod_{i=1}^d p_0^{(i)}(x_i)$ is the tensor product of 1-d distributions $p_0^{(i)}(x_i) \\propto e^{g(x_i)} \\gamma_1(dx_i)$. Then it is reasonable to expect $f(x) = \\sum_{i=1}^d g(x_i)$ can have $\\|f\\|\_{\\infty} = \\Omega(d) \\|g\\|_{\\infty}$ if the support is not restricted to a low-dimensional subspace. Because of this, it's not clear to me that the stated bounds can actually 'escape the curse of dimensionalty' or be 'dimension free' in non-trivial cases.
- Likely suboptimal bounds. There are many techniques in the literature on empirical process theory that could be used to improve the stated bounds (notably localization methods). I am not an expert on these techniques, but it would at least be worth commenting on ways the bounds in this work could be improved.
- No lower bound. It would also be helpful to prove (or at least discuss) lower bounds for this estimation problem. For example, do you know a minimax lower bound for estimating a generic function in $\\mathcal{F}\_{\\text{score}}(\\{K_i\\}_{i=1}^L)$? How does it compare to the generalization error of $\\hat{s}$? It is difficult make any takeaways from this work given that the upper bound is likely suboptimal and there is no lower bound to compare against.

Also, here are some minor typos:
- Statement of Theorem 2: please define $\hat{p}_t$. The theorem statement includes $\text{TV}(\hat{p}\_0, p\_{t\_0})$, but based on the proof (pages 31, 32) it seems like it should be $\text{TV}(\hat{p}\_{t\_0}, p\_{0})$.
- Page 7, "Proposition 3 guarantees that, by choosing $K$ ..." should be Theorem 3
- Page 7, display following "Girsanov's theorem, combined with the data processing inequality" shouldn't this be an inequality instead of an equality? Also, there seems to be inconsistent notation regarding $\hat{p}\_{t_0}$, $\hat{p}\_{t\_0, T}$, etc -- please make the notation consistent and define it.
- Page 9, first paragraph, "as the multiplication of $a_i(t)$ and $b_i(t)$ induces a nonlinearity." What is $b_i(t)$? I was not able to find any reference of it.
- Lemma 2 in the appendix is missing parenthesis in the big $O$ notation in both line items.
- Lemma 2 proof, line two of the series of inequalities (begins with $\lesssim$), this line seems to be missing a factor of $\\|f\\|\_{\mathcal{B}}$ which appears in following lines.
- Lemma 3 statement uses functions $F\_1, F\_2$ instead of $F, G$.

**Questions:**

What is the minimax rate of estimation for functions from $\\mathcal{F}\_{\\text{score}}(\\{K_i\\}_{i=1}^L)$? How does it compare to estimation rates proven in this paper? What should one takeaway from the rates in this paper, and/or what do they say about the relationship between this problem and vanilla density estimation?

---

> ### Author Response · Authors · 2023-11-16
> **Response to reviewer**
>
> We thank the reviewer for their detailed and insightful feedback. In response to the great points made, we have a few remarks:
> - It is true that the $L^{\infty}$ norm of $f$ may depend on the dimension for certain distributions. However, since the function $f$ is only defined up to an additive constant, we can replace the dependence on $\|f\|_{\infty}$ with the quantity $\frac{1}{2} \textrm{osc}(f)$, where $\textrm{osc}(f) := \sup f - \inf f$.
> - In addition, it is generally difficult to obtain sample complexity bounds where the prefactors do not depend exponentially on the dimension of the problem. In fact, in the classical nonparametric estimation of Sobolev densities, the $L^2$ error scales like $$O(\|p\|_{H^s}) n^{-\frac{s}{2s+d}}$$
>
> where the Sobolev norm can grow exponentially in $d$. For example, in the example pointed out by the reviewer where $p_0(x) = \Pi_{i=1}^d p_0^{(1)}(x_i)$,  the $H^s$  norm of $p_0$ on the unit cube $[0,1]^d$ would be $O(C^d)$ where $C \sim \|p_0^{(1)}\|_{H^s([0,1])}$, leading to an exponential dependence of $d$ when $C>1$.
>
> We stress here that our goal is to prove that, in the presence of low-complexity structure of the function $f$ defined in our paper, the **rate of convergence with respect to the growing sample size** is independent of the dimension $d$. Nonetheless, it is worth discussing the potential limitation of the constant factor $e^{\|f\|_{\infty}}$ in our bound and its comparison to classical density estimation, and we have added a remark to this end in the conclusion paper.
> - It is true that, specifically, our bound on the generalization gap (denoted $\mathcal{R}_R(\hat{\textbf{s}}) - \mathcal{R}_R(\textbf{s}^{\ast})$ in Section 4.1 the paper) is likely sub-optimal, because we bound it by the Rademacher complexity directly. We believe that a faster generalization rate can be achieved through the use of more refined techniques from empirical process theory (localization, as you point out, would probably work quite well). We have added a discussion on this in the conclusion of the paper.
> - We thank the reviewer for pointing out the very interesting question on the minimax lower bound. First we would like to clarify that $\mathcal F_{score}(\{K_i\}_{i=1}^{L})$  is class of estimators, not the target function class of the score function. Right now, we do not have a minimax lower bound for estimating densities of the form $e^{-\|x\|^2/2 + f(x)}$ with $f$ Barron, nor do we have a lower bound for estimating score functions of such distributions. Our focus here is to derive generalization error bound for the sample quality of score-based generative models instead of the estimation error of the density function itself. The main takeaway message is that the rate of the generalization error between the generated distribution and the target data distribution is dimension-free when the target density has certain low-complexity structure whereas the estimation rate of density estimation under the standard Sobolev and Holder assumptions suffers from the curse of dimensionality.
>
> We admit that the problem of obtaining a minimax lower bound of the generalization error for SGMs under our low-complexity assumption is very important and that understanding it would make the upper bound more meaningful, but it is highly non-trivial and requires deeper understanding of the approximation of the relevant multi-layer function space for the score function. We think investigation of this question is beyond the scope of the paper. We have noted this question as an interesting future direction in updated the conclusion section.
> - We thank the reviewer for catching several typos and, in response to the helpful comments on notation, we decided to begin the proof of Theorem 2 with a list of all relevant distributions and their explicit definitions. Right now, this addition is in the proof of Theorem 2 in the appendix, not the main text, due to the page limit.
>
> Note that any changes that we have made to the submitted version of the paper are colored in blue. In order to get under the page limit with the changes, we have commented out some less-important remarks, which we plan to add back in the final version.

---

### Meta-Review · Area_Chair_xxRC · 2023-12-10

**Metareview:**

The authors give the bound on the sampling error of the score-based modelling with neural networks for the certain class of starting probability distributions.

Strengths:
First paper to study sampling complexity of diffusion models.

Weaknesses:
1) The original estimate depends on the infinity norm of $\Vert f \Vert$, where $p(x) = \exp f(x) \gamma_d(dx)$, which can grow exponentially, thus saying 'break the curse' is not 100% correct. The authors claimed in their answer that this can be relaxed, for example, to Gaussian mixtures.
2) For the considered class of functions, standard Langevin samplers will also give dimension independent bounds.

**Justification For Why Not Higher Score:**

The paper is closer to borderline, due to issues mentioned in the weaknesses part.

**Justification For Why Not Lower Score:**

Although the paper has some weaknesses, it is the first paper which has such kind of theoretical study.

---

### Decision · Program_Chairs · 2024-01-16

Accept (poster)